# Set-Level Self-Supervised Learning from Noisily-Labeled Data

## Abstract

Noisy labels are inevitably presented in real-world datasets due to labeling error or visual content ambiguity. Existing methods generally approach the task of noisy label learning (NLL) by either properly regularizing the model, or reweighting clean/noisy labeled samples. While self-supervised learning (SSL) has been applied to pre-train deep neural networks without label supervision, downstream tasks like image classification still require clean labeled data. And, most SSL strategies are performed at the instance level, without accessing its label. In this paper, we propose set-level self-supervised learning (SLSSL), which performs SSL at mini-batch levels with observed noisy labels. By corrupting the labels of each training mini-batch, our SLSSL enforces the model to exhibit sufficient robustness. Moreover, the proposed SLSSL can also be utilized for sample reweighting technique. As a result, the proposed learning scheme can be applied as an expectation-maximization (EM) algorithm during model training. Extensive experiments on synthetic and real-world noisy label data confirm the effectiveness of our framework.

## 1 Introduction

Deep learning has shown tremendous success in numerous computer vision and machine learning tasks. However, collecting a large amount of precisely labeled data for training a deep neural network (DNN) is typically time-consuming and labor-intensive. Moreover, in practice, real-world datasets are usually annotated with noisy labels. In order to alleviate possible overfitting problems (Arpit et al., 2017; Zhang et al., 2017), noisy-label learning (NLL) has attracted the attention from researchers in related fields (Frenay & Verleysen, 2014; Song et al., 2021).

Recent deep-learning based NLL approaches can be categorized into two groups (Song et al., 2021; Karim et al., 2022). The first group focuses on *loss correction* (Patrini et al., 2017; Hendrycks et al., 2018; Xia et al., 2019; Wang et al., 2020; Yao et al., 2020), which learns a class-wise *noise transition matrix* to counteract the noise effect during training, so that the predicted labels can be updated accordingly. The second group of NLL works present various *sample selection* algorithms (Li et al., 2020; Nishi et al., 2021; Karim et al., 2022), aiming at filtering out noisy samples. Once the noisy labels are removed, semi-supervised learning techniques can be applied for training learning models. While promising performances have been reported, the above learning strategies rely on the prediction of the derived noise transition matrix or instance weights, which require proper learning and estimation using the training data and their noisy labels.

Instead of directly utilizing the noisy labels, self-supervised learning (SSL) has been recently applied to NLL tasks (Hendrycks et al., 2019; Ghosh & Lan, 2021; Yao et al., 2021; Ortego et al., 2021). By properly designing pretext tasks, additional supervisory signal can be derived to improve robustness of the model against label noise. However, existing SSL approaches design pretext tasks by manipulating samples at the *instance* level, regardless of the correctness of its label. As for instance-level pretext tasks (e.g., rotation prediction or contrastive-based instance discrimination), they are expected to produce proper representations from *unlabeled* data; it is not clear whether such techniques would result in robust representations when tackling the problem of noisy label learning (NLL).

In this paper, we propose a novel unique SSL approach for NLL. More precisely, we present a set-level self-supervised learning (SLSSL) strategy for training NLL models. As illustrated in Fig. 1,

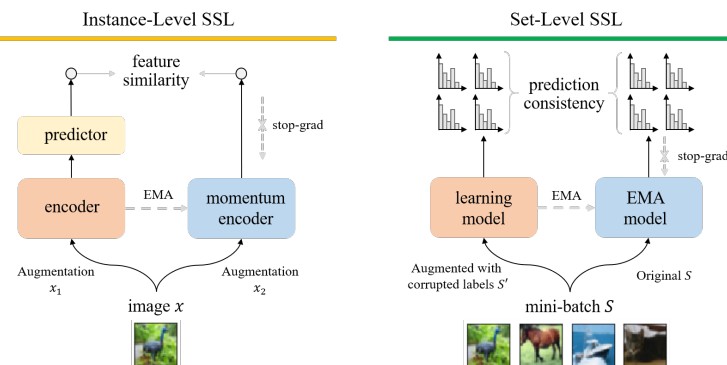

Figure 1: Illustration of our set-level self-supervised learning (SLSSL). Unlike instance-level SSL approaches, our SLSSL augments an image set (e.g., mini-batch) by manipulating its labels. By maximizing the agreement between the two augmented versions, our SLSSL results in learning models which are robust to noisily-labeled data.

given a set (mini-batch) of training samples, our SLSSL *augments* noisily labeled data by corrupting a portion of its labels for updating the DNN through a single-step optimization, while the updated model is enforced to maximize the performance agreement between different augmentation versions. As detailed later in Sect. 3, our SLSSL objective is formed to estimate the class-wise noise transition matrix, allowing us to enhance the robustness of the learned model. In addition, we show that the proposed SLSSL can be utilized to *reweight* samples for sample selection purpose. Unlike existing works that perform sample selection by assuming that instances with small losses are with clean labels, our SLSSL learns to assign larger weights to those resulting in significant performance degradation during label corruption, identifying the data with clean labels accordingly. Finally, we demonstrate that our SLSSL can be realized in an expectation-maximization (EM) like algorithm, with E-steps focusing on training model with noisy labeled data, and M-steps identifying clean data samples for training. As verified in our experiments, this alternating training strategy further boosts the performance of our framework.

The contributions of this paper are highlighted below:

- We propose set-level self-supervised learning (SLSSL) to tackle noisy-label learning (NLL) tasks, which augment image sets and enforce the model to be robust to noisy labels.

- By systematically corrupting the labels during training and enforcing prediction consistency between associated models, our SLSSL can be applied to estimate the noise transition matrix, which introduces sufficient robustness to the learned model against noisy labels.

- Our SLSSL can be further utilized to identify the label quality of each training sample, and thus sample selection for NLL can be performed accordingly.

- Our proposed learning strategy can be further viewed as an EM-like algorithm, which alternates between model training and sample reweighting for improved NLL.

## 2 RELATED WORKS

**Loss Correction for NLL**  A number of NLL works (Goldberger & Ben-Reuven, 2017; Patrini et al., 2017; Hendrycks et al., 2018; Xia et al., 2019; Wang et al., 2020; Yao et al., 2020; Zhu et al., 2022) focus on estimating the class-wise noise transition matrix of noisy training data, which describes the relationships between noisy labels and their ground-truth ones and thus can be applied to refine the predicted outputs accordingly. It is shown in (Patrini et al., 2017) that minimizing such corrected loss toward noisy labels is equivalent to optimizing the DNN toward the ground-truth labels. However, despite of their theoretical foundation, how to accurately estimate the noise transition matrix remains a challenging problem, especially when no clean training/validation sets are available.

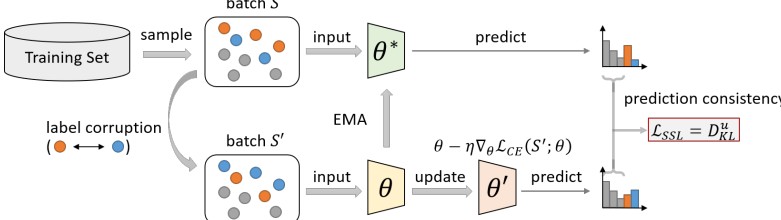

Figure 2: Illustration of our set-level self-supervised learning (SLSSL). Given a training mini-batch $S$, we first *augment* it by corrupting its labels and then obtain the *augmented* model parameter $\theta'$. We then enforce the consistency between the outputs from $f(\cdot; \theta')$ and that from the weight-averaged model $f(\cdot; \theta^*)$ to encourage model robustness against label noise.

**Sample Selection for NLL**    Selecting clean training samples according to their label confidences is another alternative for solving NLL tasks. In co-training based approaches (Malach & Shalev-Shwartz, 2017; Jiang et al., 2018; Han et al., 2018; Yu et al., 2019; Wei et al., 2020), two DNNs are trained in a collaborative manner, with each model being trained using clean samples selected by its peer model. More recently, (Li et al., 2020; Nishi et al., 2021; Karim et al., 2022; Wang et al., 2021) treat the selected clean and noisy samples as labeled and unlabeled data, respectively. They then leverage state-of-the-art semi-supervised learning techniques for training the learning models. While achieving promising results, most of the above methods perform sample selection based on the assumption that samples with small classification losses are with high probabilities for the assigned labels. This might not always holds for hard samples with correct samples to be recognized.

**Self-Supervised Learning for NLL**    Self-supervised learning (SSL) has been recently considered for approaching NLL tasks (Hendrycks et al., 2019; Ghosh & Lan, 2021; Yao et al., 2021; Ortego et al., 2021). Most existing works adapt pretext tasks that are designed for SSL on *unlabeled* data, such as predicting image rotations or contrastive-based instance discrimination, without utilizing the structural information contained in noisy labels for advanced self-supervisory signals for NLL. Recently, (Li et al., 2019) propose Meta-Learning based Noise-Tolerant (MLNT) training, in which the noisy labels are further corrupted multiple times, and a consistency constraint is designed to regularize the model toward robustness under such label corruptions. This consistency constraint can be regarded as a self-supervisory signal that takes noisy labels into account. However, as MLNT chooses to corrupt labels *randomly*, the pretext task is limited to model regularization, not able to identify data with noisy labels as sample selection methods do. As introduced in the next section, our proposed strategy can be applied for improving model robustness and estimating label confidence (i.e., sample reweighting), and our experimental results would verify the effectiveness and applicability of our approach.

## 3 PROPOSED METHOD

### 3.1 PROBLEM DEFINITION

Suppose that we are given a training set $\mathcal{D} = \{(x_n, \tilde{y}_n)\}$, with $x_n$ denoting the $n$-th image from the image space $\mathcal{X}$, and $\tilde{y}_n \in \{1, 2, ..., C\}$ as the associated (noisy) class label, which may not be necessarily consistent with its ground-truth label $y_n$. Let $f(\cdot; \theta)$ denote the NLL model parameterized by $\theta$. Our goal is to derive a robust NLL model $\theta^*$, which is derived by self-ensemble of $\theta$ using exponential moving average (EMA): $\theta^* \leftarrow \alpha\theta^* + (1 - \alpha)\theta$, with the hyperparameter $\alpha$ controlling the update speed. With model $\theta^*$ obtained, classification of test data can be performed accordingly.

### 3.2 SET-LEVEL SELF-SUPERVISED LEARNING (SLSSL)

Given a set of training samples (i.e., a mini-batch) denoted by $S = \{(x_n, \tilde{y}_n)\}_{n=1}^N$ with size $N$, we propose to *augment* it by randomly selecting two classes $i \neq j$ from $\{1, 2, ..., C\}$, followed by relabeling all the samples from label $i$ using label $j$, as illustrated in Fig. 2. This label-corrupted mini-batch is thus denoted by $S' = \{(x_n, \tilde{y}'_n)\}_{n=1}^N$, where

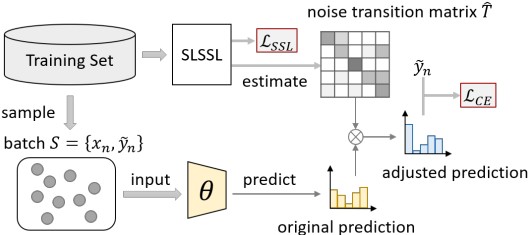

Figure 3: SLSSL for model training. Given a noisy dataset $D$, we apply SLSSL with self-supervised consistency loss $\mathcal{L}_{SSL}$ for learning robust models. The observed consistency statistics are further translated into the noise transition matrix, which refines classification prediciton accordingly.

$$\tilde{y}'_n = \begin{cases} j, & \text{if } \tilde{y}_n = i; \\ \tilde{y}_n, & \text{otherwise.} \end{cases} \tag{1}$$

We apply each augmented $S'$ to update the model $\theta$ by a single-step gradient descent using the cross-entropy classification loss. That is, we obtain $\theta'$ by:

$$\theta' = \theta - \eta \nabla_\theta \sum_{n=1}^{N} \ell_{CE}\left(f\left(x_n; \theta\right), \tilde{y}'_n\right), \tag{2}$$

where $\eta$ represents the learning rate used in such temporary updates, and $\ell_{CE}$ denotes the cross entropy loss.

By repeating the above procedure $M$ times, we obtain $M$ different models from different label corruptions of the same mini-batch, i.e., we observe $\{\theta'_m\}_{m=1}^{M}$, each corresponding to a specific label corruption between labels $(i, j)$. This set of *augmented* models $\{\theta'_m\}_{m=1}^{M}$ would guide the learning of $\theta$ and $\theta^*$ in terms of both model training and sample reweighting, as we detail in Sect. 3.3 and Sect. 3.4, respectively.

### 3.3 Set-Level SSL for Model Training

**Enforcing model robustness against label corruption**  To make our NLL model $\theta$ (and hence $\theta^*$) robust against noisy labeling, we enforce each augmented model $\{\theta'_m\}_{m=1}^{M}$ to give consistent prediction outputs on the same input $x_n$, which should be close to the output predicted by $\theta^*$. As depicted in Fig. 2, we derive each $\theta'_m$ by updating $\theta$ on a mini-batch with two classes $i$ and $j$ being corrupted (say, $i$ represents *bird* and $j$ represents *airplane*), the predictions of $f(\cdot; \theta'_m)$ for the remaining classes (e.g., *cars*) are *not* expected to be altered by this label corruption. In other words, prediction outputs for classes other than $i$ and $j$ are expected to be invariant to the above corruption. To achieve this objective, we modify the widely-used Kullback-Leibler (KL) divergence to define our SSL objective as follows:

$$\mathcal{L}_{SSL} = \frac{1}{MN} \sum_{m=1}^{M} \sum_{n=1}^{N} D_{KL}^u(f(x_n; \theta^*) \parallel f(x_n; \theta'_m)), \tag{3}$$

where $D_{KL}^u$ represents the KL divergence computed by excluding the class dimensions $i$ and $j$, which are corrupted for obtaining $\theta'_m$, i.e.,

$$D_{KL}^u(f(x_n; \theta^*) \parallel f(x_n; \theta'_m)) = \sum_{c \neq i,j} f_c(x_n; \theta^*) \log(f_c(x_n; \theta^*)/f_c(x_n; \theta'_m)). \tag{4}$$

By optimizing the SSL loss defined in equation 3, the models $f(\cdot; \theta)$ and $f(\cdot; \theta^*)$ would be encouraged to be robust to label corruption, and hence the learned model would be expected to generalize to test set data.

**Estimating noise transition matrix for NLL**   While the above SSL strategy can be viewed as a model regularization technique, previous works like (Patrini et al., 2017; Hendrycks et al., 2018; Xia et al., 2019; Wang et al., 2020; Yao et al., 2020) further deal with NLL by estimating a class-wise $C$-by-$C$ noise transition matrix $T$ for refining the classificaiton outputs.

To be more specific, this transition matrix specifies the relationships between noisy labels $\{\tilde{y}_n\}$ and ground-truth labels $\{y_n\}$, with the $(i,j)$-th entry defined by $T_{ij} = P(\tilde{y}_n = j | y_n = i)$, i.e., the probability of a sample with ground-truth label $i$ being mislabelled to label $j$. By estimating $T$ and multiplying it with the class prediction output, losses calculated for classes with corrupted labels will be suppressed. In other words, by introducing such a transition matrix in to NLL models, the final prediction output will be encouraged to be aligned with the ground-truth label.

Unfortunately, estimating the above noise transition matrix $T$ is a challenging task, especially when no clean training/validation sets are available (as the setting of our work). We now explain how we solve this task using the above set-level SSL strategy. Suppose that $\theta'_m$ is derived by updating $\theta$ on a mini-batch with classes $(i,j)$ being corrupted, as described in Sect. 3.2. Intuitively, if the samples with true class $i$ in the training dataset have a larger portion being mislabeled as $j$ (say, $i$ represents *bird* and $j$ denotes *airplane*), the model $\theta$ (and hence $\theta^*$) trained on this dataset would have biased performance towards these two categories. Thus, the performance of $\theta'_m$ would be *less* sensitive to label corruption between these two classes. On the other hand, for the case where $i$ and $j$ are less likely to be confused (say, $i$ represents *bird* and $j$ represents *truck*), the associated $\theta'_m$ would be expected to exhibit larger performance deviation.

With the aforementioned observation and property, we propose to estimate $T$ by measuring the *deviation* between $\theta'_m$ and $\theta^*$. Specifically, we adapt the standard KL divergence as:

$$D_{KL}(f(x_n; \theta^*) \parallel f(x_n; \theta'_m)) = \sum_{c=1}^{C} f_c(x_n; \theta^*) \log(f_c(x_n; \theta^*)/f_c(x_n; \theta'_m)). \tag{5}$$

To estimate the $(i,j)$-th entry $T_{ij}$, we collect all $\theta'_m$'s that are derived by corrupting $(i,j)$ across multiple mini-batches, and compute the *inverse* of the average KL divergence as:

$$Q_{i,j} = \mathbb{E}_{(i,j)} \left[ D_{KL}(f(x_n; \theta^*) \parallel f(x_n; \theta'_m)) \right]^{-\tau}, \tag{6}$$

where $\mathbb{E}_{(i,j)}$ indicates that the average is computed over all $\theta'_m$'s derived by corrupting $(i,j)$, and $\tau$ is a sharpening parameter. We then normalize $Q_{i,j}$ to obtain $\hat{T}_{ij} = Q_{i,j} / \sum_{c=1}^{C} Q_{i,c}$.

From the above objective and derivation, we are able to estimate $T$ in a per-epoch basis. We can update the matrix via EMA: let $\hat{T}_e$ denotes the matrix estimated within each epoch, and we update it by $\hat{T} = \beta\hat{T} + (1 - \beta)\hat{T}_e$, with $\beta \in [0, 1]$ controlling the update speed. As illustrated in Fig. 3, we follow forward loss correction methods (Patrini et al., 2017; Wang et al., 2020) and transform the model prediction output $f(x_n; \theta)$ by multiplying it with $\hat{T}$. This transformed prediction thus is applied for calculating the cross-entropy loss with respect to the noisy labels $\{\tilde{y}_n\}$. And, the *corrected* version of cross-entropy loss can then be expressed as follows:

$$\mathcal{L}_{CE} \triangleq -\frac{1}{N} \sum_{n=1}^{N} \tilde{y}_n \cdot \log(\hat{T}^\mathsf{T} f(x_n; \theta)), \tag{7}$$

where the superscript $\mathsf{T}$ means matrix transpose. Finally, the total loss for model training is derived by combining $\mathcal{L}_{SSL}$ in equation 3 and $\mathcal{L}_{CE}$ in equation 7.

### 3.4   SET-LEVEL SSL FOR SAMPLE REWEIGHTING

As discussed in Sect. 2, another group of NLL works choose to reweight training samples based on their labeling confidence for training the learning model. We now explain how our SLSSL can also be utilized for sample reweighting as well.

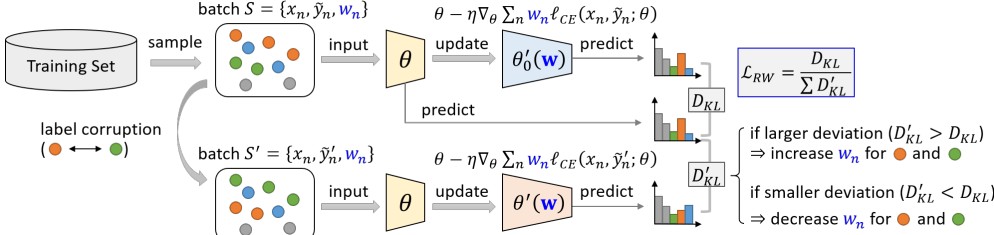

Figure 4: SLSSL for sample reweighting. Given a mini-batch $S$, we derive $\theta_0'(\mathbf{w})$ and $\theta'(\mathbf{w})$ using $S$ and its label-corrupted version $S'$, respectively. Note that *large* deviation between $\theta$ and $\theta'(\mathbf{w})$ would be observed, if such corruption is applied to clean data (when comparing to deviation between $\theta$ and $\theta_0'(\mathbf{w})$). This learning strategy allows us to increase the weights for clean training samples (and vice versa). See Sect. 3.4 for details.

As illustrated in Fig. 4, suppose that we are given a mini-batch of training samples $S = \{(x_n, \tilde{y}_n, w_n)\}_{n=1}^N$, where $w_n$ denotes the weight of the $n$-th sample $(x_n, \tilde{y}_n)$. For sample reweighting, we keep the model parameter $\theta$ fixed and only update the *sample weights* $\{w_n\}$. Recall that the estimated noise transition matrix $\hat{T}$ can be applied for transforming the model prediction in NLL. Intuitively, $\hat{T}$ should be close to an *identity* matrix if most of the samples are correctly labeled (that is, correctly-labeled samples are assigned with larger weights than those of noisily-labeled ones). In other words, the optimal weights in $S$ should make $\hat{T}_{ij}$ close to 0 for $i \neq j$ (i.e., the *off-diagonal* entries) and 1 for $i = j$ (i.e., the *diagonal* entries).

To achieve this, we first use $S$ to estimate the noise transition matrix as described in Sect. 3.3 with $\{w_n\}$ included in the estimation process, and then optimize $\{w_n\}$ based on the estimation results. Specifically, we follow the SLSSL procedure in Sect. 3.2 to create an augmented model set $\{\theta_m'(\mathbf{w})\}_{m=1}^M$, with each $\theta_m'(\mathbf{w})$ derived by updating $\theta$ using the *weighted* version of equation 2:

$$\theta_m'(\mathbf{w}) = \theta - \eta \nabla_\theta \sum_{n=1}^N w_n \ell_{CE}\left(f\left(x_n; \theta\right), \tilde{y}_n'\right), \tag{8}$$

where $\mathbf{w}$ denotes the collection of sample weights $\{w_n\}$ within the mini-batch, and $\tilde{y}_n'$ is obtained by corrupting a label pair $i \neq j$ using equation 1. To estimate $\hat{T}$, we follow Sect. 3.3 and compute the deviation between $\theta_m'(\mathbf{w})$ and $\theta$ by calculating the standard KL divergence $D_{KL}(\theta \parallel \theta_m'(\mathbf{w}))$ using equation 5. Since $\theta_m'(\mathbf{w})$ is derived by corrupting a label pair $i \neq j$, this deviation thus corresponds to the off-diagonal entry $\hat{T}_{ij}$. We further let $\theta_0'(\mathbf{w})$ denote the model derived by updating $\theta$ through a single-step gradient descent based on the *original* mini-batch $S$ (without label corruption), and we also compute its deviation from $\theta$ (denoted by $D_{KL}(\theta \parallel \theta_0'(\mathbf{w}))$) to represent all diagonal entries of $\hat{T}$. Since we estimate $\hat{T}$ by taking inverse of the above deviations, for $\hat{T}$ to be close to identity, $D_{KL}(\theta \parallel \theta_m'(\mathbf{w}))$ should be much larger than $D_{KL}(\theta \parallel \theta_0'(\mathbf{w}))$. Thus, our sample reweighting (RW) loss can be defined in the following contrastive form as:

$$\mathcal{L}_{RW} = \frac{D_{KL}(\theta \parallel \theta_0'(\mathbf{w}))}{\sum_{m=1}^M D_{KL}(\theta \parallel \theta_m'(\mathbf{w}))}. \tag{9}$$

It can be seen that, the denominator in $\mathcal{L}_{RW}$ describes the performance deviations between the original model and their augmented versions derived by multiple label corruptions. For a mini-batch with more clean samples, such label corruptions would lead to larger deviations, as compared to the case without label corruption (i.e., the numerator). We thus propose to minimize $\mathcal{L}_{RW}$ for sample reweighting. In other words, samples leading to higher performance deviations from label corruptions are thus assigned with larger confidences/weights. This serves as our sample reweighting strategy based on SLSSL.

Table 1: Classification accuracy (%) on CIFAR-10 across different noisy labeling schemes with NLL methods based on a single learning model.

| Method | No noise | Sym 50% | Sym 70% | Asym 30% | Asym 40% |
|---|---|---|---|---|---|
| Baseline (cross-entropy) | 92.53±0.07 | 79.87±0.29 | 70.93±0.38 | 86.17±0.17 | 83.95±0.36 |
| F-correction (Patrini et al., 2017) | 92.63±0.12 | 81.22±0.04 | 73.74±0.46 | 88.28±0.23 | 87.45±0.24 |
| MLNT (Li et al., 2019) | 93.76±0.03 | 87.18±0.11 | 80.76±0.29 | 91.84±0.28 | 89.57±0.12 |
| Our SLSSL (E-step only) | 93.86±0.08 | 89.11±0.08 | 81.22±0.42 | 92.67±0.06 | 91.12±0.10 |
| Our SLSSL (EM) | **94.17±0.02** | **90.13±0.16** | **83.91±0.30** | **92.94±0.16** | **91.78±0.12** |

## 3.5 SLSSL AS AN EM-LIKE ALGORITHM

It is worth noting that, we can further integrate the proposed model training (Sect. 3.3) and sample reweighting (Sect. 3.4) schemes as an EM-like algorithm for NLL. To be more specific, the E-step focuses on model training with sample weights being fixed, while the M-steps aims to reweight training samples with the derived model. In practice, we first randomly initialize $\theta$ and train it from scratch by optimizing $\mathcal{L}_{SSL}$ in equation 3 and $\mathcal{L}_{CE}$ in equation 7 on the unweighted training dataset ($w_n = 1$ for all $n$). After obtaining the best model through validation (usually the weight-averaged model $\theta^*$), we then fix its parameters and utilize it to optimize all $\mathbf{w}$ by optimizing equation 9. The updated sample weights are then applied in the model training again. As confirmed by our experiment, this alternative optimization strategy would further boost the NLL performance.

## 4 EXPERIMENTS

## 4.1 DATASETS AND SETTINGS

**Datasets**   We conduct experiments on the datasets of CIFAR-10 (Krizhevsky & Hinton, 2009), CIFAR-10N (Wei et al., 2022), and Clothing1M (Xiao et al., 2015). The CIFAR-10 dataset contains 50K training images and 10K test images from 10 categories. We follow (Li et al., 2019; 2020) to corrupt the training labels by two types of noise: *symmetric* and *asymmetric*. For symmetric noise, each sample has a fixed probability to be labeled uniformly into other classes. For asymmetric noise, 5 transition patterns (*bird* $\rightarrow$ *airplane*, *cat* $\leftrightarrow$ *dog*, etc.) are proposed to simulate the class-dependent noise. Fig. 5(a) shows the transition matrix used to construct a noisy CIFAR-10 dataset with 40% asymmetric noise. CIFAR-10N is extended CIFAR-10, with each training image containing three human-annotated labels (denoted as *Random-i*, $i \in \{1, 2, 3\}$). The three noisy labels for each image are further aggregated by majority voting (denoted as *Aggregate*) and randomly picking one wrong label if any (denoted as *Worst*). Clothing1M consists of 1M training images from 14 categories of clothes. The labels are extracted from the surrounding texts of images and are thus practically noisy. We use the 14K clean validation set and the 10K test set.

**Implementation details**   For both CIFAR-10 and CIFAR-10N, we follow DivideMix (Li et al., 2020) to adopt a Pre-Act ResNet-18 network (He et al., 2016) (and its hyperparameters). We also integrate our SLSSL sample reweighting scheme (i.e., the M-step in Sect. 3.4) into the *co-dividing* step of DivideMix for evaluation. For Clothing1M, we again follow DivideMix and use an ImageNet-pretrained ResNet-50 network. To compare with recent state-of-the-art methods, we reproduce the experiment results for DivideMix on all the three datasets, as well as Forward Loss Correction (F-correction) (Patrini et al., 2017) and MLNT (Li et al., 2019) on CIFAR-10. Please refer to Appendix A for the more details.

## 4.2 QUANTITATIVE RESULTS

**Comparisons with single-model approaches**   We first compare the proposed SLSSL with single-model NLL approaches using the same Pre-Act ResNet-32 backbone, including F-correction (Patrini et al., 2017) and MLNT (Li et al., 2019). The performances are summarized in Table 1. Here, we follow the same procedure in MLNT to generate symmetric and asymmetric noisy labels from CIFAR-10, and report the mean and standard error values of test accuracy across 3 runs. As can be seen from the table, with SLSSL-based model training along (i.e., E-step in Sect. 3.3), our framework achieved significant performance improvement across all noise ratios. By further applying

Table 2: Classification accuracy (%) on CIFAR-10 across different noisy labeling schemes with NLL methods adopting dual models (i.e., co-training based approaches).

| Noise type | | Sym | | | | Asym | Mean |
|---|---|---|---|---|---|---|---|
| Method/Noise rate | | 20% | 50% | 80% | 90% | 40% | |
| DivideMix (Li et al., 2020) | Best | 96.1 | 94.6 | 93.2 | 76.0 | 93.4 | 90.66 |
| | Last | 95.7 | 94.4 | 92.9 | 75.4 | 92.1 | 90.10 |
| DM-AugDesc (Nishi et al., 2021) | Best | 96.3 | 95.6 | 93.7 | 35.3 | 94.4 | 83.06 |
| | Last | 96.2 | 95.4 | 93.6 | 10.0 | 94.1 | 77.86 |
| SLSSL (E-step only, single-model) | Best | 93.5 | 90.3 | 76.4 | 57.3 | 92.3 | 81.96 |
| | Last | 93.1 | 90.1 | 76.1 | 57.0 | 91.8 | 81.62 |
| SLSSL (EM single-model) | Best | 95.8 | 94.4 | 93.2 | 67.0 | 93.1 | 88.70 |
| | Last | 95.2 | 94.0 | 92.8 | 61.5 | 92.4 | 87.18 |
| SLSSL (EM dual-model) | Best | 96.3 | 95.0 | 93.4 | 77.5 | 94.2 | **91.28** |
| | Last | 96.2 | 94.8 | 93.2 | 76.7 | 93.9 | **90.96** |

Table 3: Classification accuracy (%) on CIFAR-10N with different human-labeling schemes.

| Method/Noise type | | *Aggregate* | *Random*-1 | *Random*-2 | *Random*-3 | *Worst* | Mean |
|---|---|---|---|---|---|---|---|
| DivideMix (Li et al., 2020) | Best | 95.2 | 95.5 | 95.5 | 95.5 | 93.0 | 94.94 |
| | Last | 95.0 | 95.1 | 95.2 | 95.2 | 92.6 | 94.62 |
| Our SLSSL | Best | 95.7 | 95.5 | 95.8 | 95.3 | 93.3 | **95.12** |
| | Last | 95.6 | 95.3 | 95.5 | 95.1 | 93.1 | **94.92** |

sample reweighting (i.e., M-step in Sect. 3.4), the performances were further improved by a large margin, which confirms the effectiveness of our proposed SLSSL as a sample reweighting strategy.

**Comparisons with state-of-the-art dual-model approaches**   Next, we compare the performance of SLSSL with recent state-of-the-art methods utilizing two networks in the co-training fashion, including DivideMix (Li et al., 2020), and DM-AugDesc (Nishi et al., 2021). Table 2 lists the performance comparisons on the synthetic noisy datasets generated from CIFAR-10. Following DivideMix and DM-AugDesc, we report both the *best* test accuracy across all training epochs, and the average test accuracy over the *last* 10 epochs. As can be seen from the table, our frameworks performed favorably against state-of-the-art methods, including DivideMix on which we apply our SLSSL. It is worth noting that, DM-AugDesc also adapted DivideMix and focused on searching for the *best* augmentation strategies for NLL. On the other hand, we only apply standard augmentation techniques such as random cropping and horizontal flipping in our experiments.

Table 3 lists the performance comparisons between SLSSL and DivideMix (Li et al., 2020) on the CIFAR-10N dataset. It can be seen that our method outperformed DivideMix on this challenging human-annotated dataset. Table 4 further presents results on the Clothing1M dataset, which confirms the effectiveness of our proposed SLSSL over state-of-the-art methods.

### 4.3 VISUALIZATION AND ANALYSIS

**Noise transition matrix for NLL**   We show the noise transition matrix $\hat{T}$ estimated based on Sec. 3.3 on the CIFAR-10 dataset with $40\%$ asymmetric noise. We first trained a model by applying our E-step for 100 epochs, and then conducted the M-step for 20 epochs to assign weights for all training samples. Based on the sample weights, we then trained our model again using E-step for 100 epochs. As can be seen from Fig. 5(b), the noise transition matrix estimated after the first E-step is fairly close to the ground-truth showed in Fig. 5(a). In Fig. 5(c), on the other hand, the estimated noise transition matrix becomes much closer to an identity matrix after a complete E-M-E cycle, indicating the effectiveness of our M-step in assigning sample weights.

**Reweighted samples**   Finally, we show the sample weights optimized based on our SLSSL at the end of three successive M-steps on CIFAR-10 with $40\%$ asymmetric noise. As can be seen from Fig. 6, as the optimization progresses, the sample weights of clean and noisy samples become increasingly separable (with AUC scores reported in the figure). This further confirms the effectiveness of our SLSSL for sample reweighting in NLL.

Table 4: Classification accuracy (%) on Clothing1M. Note that DM-AugDesc (Nishi et al., 2021) is designed to search for best augmentation techniques for NLL, while we simply perform random cropping and flipping in SLSSL.

| Method | Test Accuracy |
|---|---|
| Cross-Entropy | 69.21 |
| F-correction (Patrini et al., 2017) | 69.84 |
| MLC (Wang et al., 2020) | 71.06 |
| MLNT (Li et al., 2019) | 73.47 |
| T-Revision (Li et al., 2020) | 74.18 |
| DivideMix (Li et al., 2020) | 74.76 |
| DM-AugDesc (Nishi et al., 2021) | 75.11 |
| Jo-SRC (Yao et al., 2021) | **75.93** |
| DivideMix (reproduced) | 74.34 |
| Our SLSSL | **74.51** |

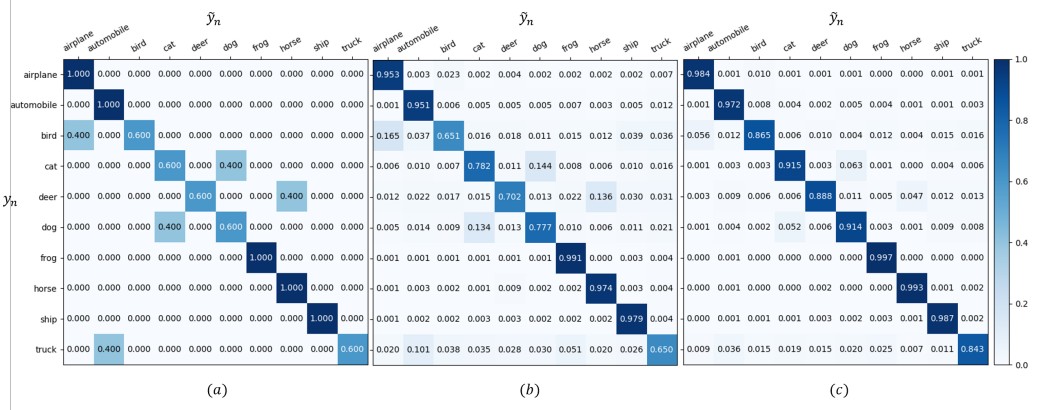

Figure 5: (a) The ground-truth transition matrix for constructing the noisy label dataset with 40% asymmetric noise from CIFAR-10; (b) our estimated matrix after the first E-step, which is close to the ground truth; (c) our estimated matrix after a complete E-M-E cycle (which is close to the identity matrix, implying that our SLSSL assigns proper sample weights for cleaning the training data).

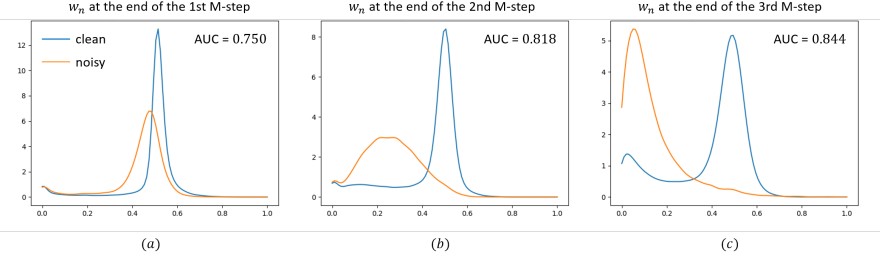

Figure 6: Empirical distributions of sample weights derived at the end of three consecutive M-steps (from (a)-(c)) for CIFAR-10 with 40% asymmetric noise. Note that AUC scores are shown in each sub-figure, quantitatively verifying our separation between clean and noisy labeled data.

## 5 CONCLUSION

In this paper, we proposed set-level self-supervised learning (SLSSL) to address noisy label learning (NLL) tasks. SLSSL performs self-supervised learning at mini-batch levels without prior knowledge on noisy label distribution. By corrupting the labels of each training mini-batch and applying corresponding consistency constraints, our SLSSL enforces the model to exhibit sufficient robustness toward labeling noise. Moreover, the proposed SLSSL can also be utilized for sample reweighting technique based on a novel high-impact principle. Finally, we demonstrate that the proposed learning scheme can be viewed as an expectation-maximization (EM) algorithm for training NLL models. With experiments conducted on synthetic (CIFAR-10) and real-world (CIFAR-10N and Clothing1M) noisy label data, the effectiveness of our proposed SLSSL can be sucessfully verified.

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

## A    IMPLEMENTATION DETAILS

**Single-model experiment on CIFAR-10**    For CIFAR-10, we follow previous works (Tanaka et al., 2018; Li et al., 2019) and split the 50K training samples into 45K training and 5K validation subsets. We then introduce symmetric or asymmetric label noise to the training subset as described in Sec. 4.1. All models are trained on the noisy training subset, with hyperparameters being chosen based on the model performance on the clean validation subset. After hyperparameter tuning, we then introduce label noises to all 50K training samples and rerun all experiments for comparison.

We develop our SLSSL algorithm based on the implementation of MLNT[1] (Li et al., 2019), which adopts a Pre-Act ResNet-32 network. We follow most of the hyperparameters from MLNT, and train the model for 300 epochs using SGD with an initial learning rate 0.02 (divided by 2 after every 50 epochs), a momentum of 0.9, a weight decay of 0.0005, and a batch size of 128. The learning rate $\eta$ used for obtaining the augmented models is set to 0.1 in equation 2. For our SLSSL framework with both E and M steps, we set $\eta = 0.001$ in equation 8, and conduct the M-step for 5 iterations (3 M-step epochs per iteration) at every 50 E-step model training epochs: $\{50, 100, 150, 200, 250\}$.

As for other single-model methods, we simply train the model for 300 epochs using standard cross-entropy classification loss as the Baseline, and add the ground-truth noise transition matrix for F-correction (Patrini et al., 2017). The MLNT results are reproduced by directly using their implementation.

**Dual-Model co-training for CIFAR-10 and CIFAR-10N**    We follow the same principle from the above single-model experiment on CIFAR-10 for hyperparameter tuning, and directly apply the best set of hyperparameters for the 40% asymmetric noisy dataset to CIFAR-10N.

We integrate our SLSSL algorithm into the co-training phase of DivideMix (Li et al., 2020) based their implementation[2], and also follow most of their hyperparameters. We first train two Pre-Act ResNet-18 networks separately for 10 epochs in the warm-up phase, and then enter the co-training phase for the remaining 290 epochs. At the 150 epoch, we replace DivideMix's GMM-based reweighting procedure on the 2nd network by our SLSSL-based sample reweighting algorithm, and conduct the M-step for 60 iterations (2 M-step epochs per iteration) at every 5 model training epochs: $\{150, 155, ..., 295\}$. Based on the derived sample weights, we then follow standard DivideMix and apply *co-dividing*, *co-refinement*, and *co-guessing* techniques to train the two networks.

**Dual-Model co-training for Clothing1M**    For Clothing1M, we also follow DivideMix (Li et al., 2020) and train our model on the 1M (noisy) training dataset, with hyperparameters being chosen based on the 14K (clean) validation subset. Similar to CIFAR-10, we integrate our SLSSL algorithm into the co-training phase of DivideMix. We start our SLSSL-based sample reweighting (i.e., the M-step in Sect. 3.4) at epoch 70. Since only 32K samples are randomly selected per training epoch, we conduct M-step for every epoch, and set the total epoch number in the co-training phase as 80. The learning rate is set to 0.002, and decays with a factor of 10 after the epoch 40. All the rest hyperparameters follow DivideMix.

**Algorithm**    We provide the pseudo codes for our SLSSL in Algorithm 1 for model training ( 3.3; E-step) and Algorithm 2 for sample reweighting ( 3.4; M-step). The two algorithms can be combined as an EM-like iterative training strategy as described in Sect. 3.5.

## B    VISUALIZATION ON CIFAR-10

We provide additional training statistics to further validate the proposed methods. In Fig. 7(a), we show the Area Under the Curve (AUC) for clean/noisy sample classification on CIFAR-10 with 40% asymmetric label noise in the single-model experiment. As can be seen, our M-step is able to effectively identify clean/noisy samples across successive M-steps, even for *bird* samples with 40% samples being mislabeled as *airplane*, and for *cat* samples which could be heavily confused with *dog* samples with 40% samples in each class being mislabeled as one another. In Fig. 7(b), we

---

[1]https://github.com/LiJunnan1992/MLNT

[2]https://github.com/LiJunnan1992/DivideMix

---

**Algorithm 1** SLSSL-E (model training in Sect. 3.3)

---

1: **Input**: Noisy training dataset $\mathcal{D} = \{(x_n, \tilde{y}_n, w_n)\}$ (fixed $\{w_n\}$), learning model $\theta$, EMA model $\theta^*$, number of E-step epochs $P$, number of label corruption for each mini-batch $M$, estimated noise transition matrix at the $e^{th}$ epoch $\hat{T}_e$, estimated noise transition matrix $\hat{T}$.
2: **if** first E-step **then**
3:     Random initialize $\theta$
4:     Initialize EMA model $\theta^* = \theta$
5:     Initialize $\hat{T}$ as identity
6:     Initialize all sample weights $w_n = 1$
7: **end if**
8: **for** $e = 1$ to $P$ **do**
9:     **while** not done **do**
10:         Sample mini-batch $S = \{(x_n, \tilde{y}_n, w_n)\}_{n=1}^N$ from $\mathcal{D}$
11:         **for** $m = 1$ to $M$ **do**
12:             Randomly sample a class pair $(i, j)$
13:             Obtain augmented set $S' = \{(x_n, \tilde{y}'_n, w_n)\}_{n=1}^N$ by corrupting $(i, j)$ in $S$ (Eq. 1)
14:             Derive augmented model $\theta'_m$ from $\theta$ by single-step gradient descent on $S'$ (Eq. 2)
15:             Compute the KL divergence $D_{KL}$ between $\theta'_m$ and $\theta^*$ (Eq. 4)
16:         **end for**
17:         Compute set-level self-supervised loss $L_{SSL}$ (Eq. 3)
18:         Update $\theta$ by minimizing $L_{SSL}$
19:         Compute classification loss $L_{CE}$ and then correct it using $\hat{T}$ (Eq. 7)
20:         Update $\theta$ by minimizing $L_{CE}$
21:         Aggregate $D_{KL}$'s derived by corrupting $(i, j)$ to estimate the $(i, j)^{th}$ entry of $\hat{T}_e$ (Eq. 6)
22:     **end while**
23:     Update $\hat{T}$ using EMA of $\hat{T}_e$
24:     Update $\theta^*$ using EMA of $\theta$
25: **end for**

---

**Algorithm 2** SLSSL-M (sample reweighting in Sect. 3.4)

---

1: **Input**: Noisy training dataset $\mathcal{D} = \{(x_n, \tilde{y}_n, w_n)\}$, learning model $\theta$ (fixed), number of M-step epochs $Q$, number of label corruption for each mini-batch $M$.
2: **for** $e = 1$ to $Q$ **do**
3:     **while** not done **do**
4:         Sample mini-batch $S = \{(x_n, \tilde{y}_n, w_n)\}_{n=1}^N$ from $\mathcal{D}$
5:         **for** $m = 1$ to $M$ **do**
6:             Randomly sample a class pair $(i, j)$
7:             Obtain augmented set $S' = \{(x_n, \tilde{y}'_n, w_n)\}_{n=1}^N$ by corrupting $(i, j)$ in $S$ (Eq. 1)
8:             Derive $\theta'_m(\mathbf{w})$ from $\theta$ by weighted single-step gradient descent on $S'$ (Eq. 8)
9:             Derive $\theta'_0(\mathbf{w})$ from $\theta$ by weighted single-step gradient descent on $S$ (Eq. 8)
10:         **end for**
11:         Compute the sample reweighting loss $L_{RW}$ (Eq. 9)
12:         Update $\{w_i\}_{i=1}^N$ by minimizing $L_{RW}$
13:     **end while**
14: **end for**

---

show the training and testing accuracy values. As can be seen, the accuracy is improved after each iteration of M-step at epochs 50 and above, further validating the effectiveness of our SLSSL-based sample reweighting procedure.

## C   Additional Experiment Results

**CIFAR-10 and CIFAR-10N**   In Tables 2 and 4, results of DivideMix (Li et al., 2020) and DM-AugDesc (Nishi et al., 2021) were directly copied from their papers (i.e., no standard deviation were reported by neither works). Here, we further conduct additional runs with different random seeds for our SLSSL and DivideMix on selected noisy settings from CIFAR-10 and CIFAR-10N. The results

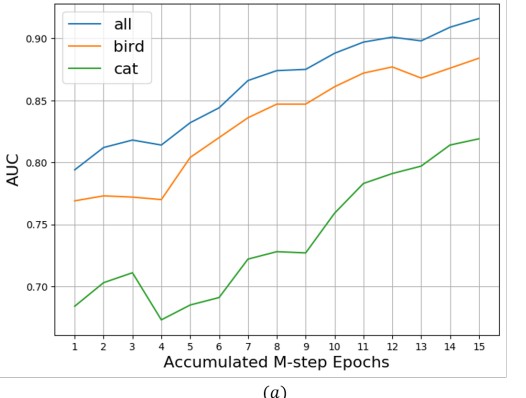 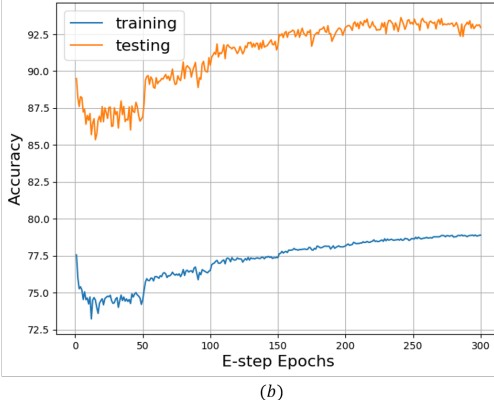

$(a)$ $(b)$

Figure 7: Evaluation on CIFAR-10 with 40% asymmetric label noise. (a) Area Under the Curve for clean/noisy sample classification based on the sample weights $\{w_n\}$ obtained by our M-steps. The X-axis indicates the *accumulated* M-step epochs over 5 iterations (3 M-step epochs per iteration) at E-step epochs from 50 up to 250. Note that our M-step is shown to identify clean/noisy samples across successive M-steps by assigning proper weights, improving both Net1 and Net2 in the co-training process. (b) Training and testing accuracy values on the first network over all 300 training epochs. The accuracy is improved after each iteration of M-step at epochs 50 and above.

Table 5: Classification accuracy (%) on CIFAR-10 across different noisy labeling schemes with NLL methods adopting dual models (i.e., co-training based approaches). We report results over 3 independent runs.

| | | CIFAR-10 | CIFAR-10N |
|---|---|---|---|
| Method | | Asym 40% | *Aggregate* |
| DivideMix (Li et al., 2020) (reproduced) | Best | 93.20±0.20 | 95.37±0.12 |
| | Last | 93.25±0.15 | 95.10±0.08 |
| Our SLSSL | Best | **94.20±0.10** | **95.73±0.12** |
| | Last | **93.75±0.15** | **95.57±0.12** |

are listed in Table 5. Following the setting of DivideMix (Li et al., 2020), we report the best test accuracy (Best) and the averaged test accuracy over the last 10 epochs (Last). From this table, it can be seen that the improvements of our SLSSL over DivideMix were statistically significant.

**CIFAR-100**  To provide additional performance evaluation, we also add experiments on CIFAR-100. We select two noise types (20% and 50% symmetric noise) and report results from a single run using the same random seed to ensure the same noise setting between DivideMix and our SLSSL. As can be seen from Table 6, our SLSSL again outperformed DivideMix and also MD-DYR-SH (Arazo et al., 2019) on CIFAR-100.

## D    ADDITIONAL COMPARISONS

**Compare to instance-level SSL approaches**  As discussed in Sect. 1, it is not clear whether existing instance-level SSL techniques would result in robust representations when tackling the NLL tasks. To make our discussions and comparisons more complete, we compare our method to MOIT (Ortego et al., 2021), a recent instance-based SSL approach to NLL, on CIFAR-10. For fair comparisons, we follow MOIT to adapt a single PreAct ResNet-18 network, and report the test accuracy in the last training epoch. As can be seen fromTable 7, our SLSSL reached comparable performance for asymetric noise at 40%, while outperformed MOIT with significant margins for different symmetric noise levels.

**Compare to recent related works**  SOP (Liu et al., 2022) proposes to model the label noise and learn to separate it from the data by enforcing its sparsity, and serves as one of the most recent works of NLL. However, as can be seen from Table 8, our SLSSL still performs favorably against SOP, especially on the more challenging Clothing1M dataset.

Table 6: Classification accuracy (%) on CIFAR-100 across different noisy labeling schemes with NLL methods adopting dual models (i.e., co-training based approaches). For DivideMix(Li et al., 2020) and our SLSSL, we report result from a single run using the same random seed.

| Method | | Sym 20% | Sym 50% |
|---|---|---|---|
| MD-DYR-SH (Arazo et al., 2019) (reported) | Best | 73.9 | 66.1 |
| | Last | 73.4 | 65.4 |
| DivideMix (Li et al., 2020) (reproduced) | Best | 77.50 | 74.20 |
| | Last | 77.00 | 73.80 |
| Our SLSSL | Best | **78.08** | **74.34** |
| | Last | **77.83** | **73.95** |

Table 7: Classification accuracy (%) on CIFAR-10 across different noisy labeling schemes from MOIT (Ortego et al., 2021) and our SLSSL.

| | Sym 20% | Sym 80% | Asym 40% | Mean |
|---|---|---|---|---|
| MOIT (Ortego et al., 2021) (reported) | 94.08 | 75.83 | 93.27 | 87.73 |
| Our SLSSL | 95.18 | 92.82 | 92.23 | **93.41** |

## E  LIMITATIONS

Since our proposed SLSSL can be viewed as a unique meta-learning scheme on existing NLL methods like DivideMix (Li et al., 2020), we expect longer computation time during training. However, take CIFAR-10 for example, DivideMix took 15 hours to train using a single Nvidia TITAN-V GPU, while the full version of our SLSSL (i.e., implemented as an EM algorithm) required 25 hours. It can be seen that, the overall computation time of our SLSSL is still in the same order of that of SOTAs like DivideMix.

Also, as noted in (Patrini et al., 2017; Hendrycks et al., 2018; Xia et al., 2019; Wang et al., 2020; Yao et al., 2020), NLL methods based on class-wise noise transition matrix estimation share the limitation that the number of classes for NLL would be reasonable (e.g., 100 in CIFAR-100). This is to avoid the potential problem of estimating a large noise transition matrix. Sharing the concern of the above works, this would also among the current limitation of our work.

Table 8: Classification accuracy (%) on CIFAR-10N (Aggregate) and Clothing1M from SOP (Liu et al., 2022) and our SLSSL.

|  | CIFAR-10N (Aggregate) | Clothing1M |
|---|---|---|
| SOP (Liu et al., 2022) (reported) | 95.61 | 73.5 |
| Our SLSSL | **95.73** | **74.51** |

