# OpenReview forum: "Set-Level Self-Supervised Learning from Noisily-Labeled Data"
_ICLR.cc/2023/Conference — Submitted to ICLR 2023_

### Official Review · Reviewer_szQw · 2022-10-15

**Confidence:** 4
**Correctness:** 3
**Technical Novelty And Significance:** 2
**Empirical Novelty And Significance:** Not applicable
**Recommendation:** 5

**Clarity, Quality, Novelty And Reproducibility:**

The paper is generally well-written. It would be better if the advantages of set-level augmentation over instance-level augmentation can be explained in a clearer way. The code is not provided.

**Strength And Weaknesses:**

Pros:

1. The idea of conducting set-level self-supervised learning for tackling label noise is novel and interesting.
2. The proposed algorithm is rigorously explained.
3. The experiments are convincing.

Cons:

1. I’m not fully understand the sentence “As for instance-level pretext tasks (e.g., rotation prediction or contrastive-based instance discrimination), they are expected to produce proper representations from unlabeled data; it is not clear whether such techniques would result in robust representations when handling input data with noisy labels” in the introduction. In fact, the pretext tasks usually do not depend on labels, so I’m not clear why the data with noisy labels might affect the model output aided by this technique. I cannot see the connections between these two things.
2. It would be better if pseudo code of the proposed SLSSL can be provided.
3. I’m not fully understand why the designed set-level augmentation and label corruption way (e.g. Eq. 1) can help resist label noise. It would be more helpful if some explanations can be provided.
4. In fact, for label noise learning, the estimation of label transition matrix is a quite challenging task. I want to see the estimation accuracy when the data dimension is high.


**Summary Of The Paper:**

This paper proposes a new set-level self-supervised learning (SLSSL) method for combating noisy labels. By corrupting the labels of each training mini-batch, the proposed SLSSL enables the model to exhibit sufficient robustness. The comparison with SOTA methods demonstrates the effectiveness of the proposed method.

**Summary Of The Review:**

I do not have major concerns on this paper. It seems that the proposed algorithm works well.

---

> ### Author Response · Authors · 2022-11-18
> **Response to Reviewer szQw (Part 3)**
>
> **Q3: I’m not fully understand why the designed set-level augmentation and label corruption way (e.g. Eq. 1) can help resist label noise. It would be more helpful if some explanations can be provided.**
>
> A3: We thank the reviewer for raising up the concerns. We now present the intuition on why our set-level label manipulation allows training NLL models to resist label noise.
>
> As pointed out in MLNT [C], the main issue for training a model directly on noisily-labeled data is that it tends to overfit to the noisy labels and thus yields poor generalization. One key approach to address this issue is to optimize the model’s parameters toward the direction of less prone to overfitting and more robust against label noise. As discussed in Q1, while instance-based SSL-based approaches to NLL like Jo-SOC [D] and MOIT [B] have been proposed, such SSL techniques did not utilize the observed noisy labels in their pretext learning stage.
>
> Our SLSSL advances label corruption for each mini-batch during training, followed by enforcing the prediction consistency across models learned from each label-corruped mini-batch. Inspired by MAML, we apply single-step gradient descent to realize the above training scheme, encouraging the learned model to exhibit sufficient robustness against label noise. In other words, with such a meta-learning strategy and consistency constraint, the learned model would become robust to noisy training dataset, and it would be expected to achieve comparable performance as the model learned from a *clean* training dataset (which cannot be possibly observed during training). We hope the above explanations, plus the connection to MAML, would address the reviewer’s concern.
>
>
> **Q4: In fact, for label noise learning, the estimation of label transition matrix is a quite challenging task. I want to see the estimation accuracy when the data dimension is high.**
>
> A4: We thank the reviewer for pointing out a practical concern. First of all, the raw input data considered in this paper are images, which typically have standard data dimension (e.g., (32, 32, 3) or (224, 224, 3) in (H, W, C) format). Secondly, the feature dimension are determined by the used feature extractor (e.g., 512 for the adapted PreAct ResNet-18). Finally, the considered datasets are medium to large-scale (e.g., 50K training images for CIFAR-10/CIFAR-10N/CIFAR-100, and 1M for Clothing1M), while the size of the transition matrix is based on the number of categories (i.e., a C x C matrix).
>
> We agree that estimation of a large transition matrix would be a very challenging task, and that is the reason why existing NLL works generally consider the above datasets. We thank the suggestion from the reviewer again, and we will list the last remark as a shared limitation of recent NLL works and as one of our future research directions.
>
> **Q5: It would be better if the advantages of set-level augmentation over instance-level augmentation can be explained in a clearer way.**
>
> A5: We thank the reviewer for raising this concern. Following the remarks in Q1, existing SSL approaches only perform instance-level data augmentation, with the corresponding objectives not necessarily related to the task of NLL. To address this concern, we propose set-level self-supervision learning (SLSSL) on noisy label data. As explained in Q1, we introduce a novel data augmentation strategy at the set level via label corruptions, which allowsself-supervision to be observed directly from noisily-labeled training data. Specifically, our SLSSL advances label corruption for each mini-batch during training, followed by enforcing the prediction consistency across the resulting models derived by single-step gradient descent using the label-corrupted mini-batches, encouraging the learned model to exhibit sufficient robustness against label noises. As discussed in Q3, such a meta-learning strategy and consistency constraint would encourage the learning model to become more tolerant against different label noises. As for the empirical support, please kindly refer to the table in Q1, which confirms the advantages of our set-level augmentation over instance-level augmentation.
>
> **Q6: The code is not provided.**
>
> A6: We thank the reviewer for the critical concern and suggestion. We will be glad to release the source code (with possible revision and additional experiments suggested by all seven reviewers).
>
> **Rerefences**
> [A] C. Finn et al., Model-agnostic meta-learning for fast adaptation of deep networks, ICML 2017
> [B] D. Ortego et al., Multi-objective interpolation training for robustness to label noise, CVPR 2021
> [C] J. Li et al., Learning to learn from noisy labeled data, CVPR 2019
> [D] Y. Yao et al., Jo-src: A contrastive approach for combating noisy labels, CVPR 2021

---

> > ### Comment · Reviewer_szQw · 2022-11-29
> > **Thanks for your response**
> >
> > Many thanks for providing the response to my previous concerns. This paper is potentially interesting. However, considering the unclear motivation as well as the comments of other reviewers, I also feel that this paper is immature to be published now, so I will lower my rating accordingly.

---

> ### Author Response · Authors · 2022-11-18
> **Response to Reviewer szQw (Part 2)**
>
> **Q2: It would be better if pseudo code of the proposed SLSSL can be provided.**
>
> A2: We thank the reviewer for this suggestion. Please see the pseudo code below, and  we will add this algorithm to the appendix.
>
> * Variables
>     * Noisy training dataset: $D$
>     * Learning model: $\theta$
>     * EMA model: $\theta^*$
>     * Number of E-step epochs: $P$
>     * Number of M-step epochs: $Q$
>     * Number of label corruption for each mini-batch: $M$
>     * Estimated noise transition matrix at the $e$-th epoch: $\hat{T}_e$
>     * Estimated noise transition matrix: $\hat{T}$
>
> * E-step (Sect. 3.3)
>     * **if** first E-step
>         * Random initialize $\theta$
>         * Initialize $\theta^*$ by $\theta$
>         * Initialize $\hat{T}$ as identity
>     * **for** $e$ in $[1, 2, ...,P]$
>         * **while** not done **do**
>             * Sample mini-batch $S$ from $D$
>             * **for** $m$ in $[1, 2, ...,M]$
>                 * Randomly sample a class pair $(i, j)$
>                 * Obtain the augmented set $S'$ by corrupting labels $(i, j)$ in $S$ (Eq. 1)
>                 * Derive the augmented model $\theta'_m$ by applying single-step gradient descent on $\theta$ using $S'$ (Eq. 2)
>                 * Compute the KL divergence between $\theta'_m$ and $\theta^*$ (Eq. 4)
>             * Compute set-level self-supervised loss $L_{SSL}$ (Eq. 3)
>             * Update theta by minimizing $L_{SSL}$
>             * Compute classification loss $L_{CE}$ and then correct it using $\hat{T}$ (Eq. 7)
>             * Update theta by minimizing $L_{CE}$
>             * Aggregate all KL divergences derived by corrupting $(i, j)$ to estimate the $(i, j)$-th entry of $\hat{T}_e$ (Eq. 6)
>         * Update $\hat{T}$ using EMA of $\hat{T}_e$
>         * Update $\theta^*$ using EMA of $\theta$
>
> * M-step (Sect. 3.4)
>     * **for** $e$ in $[1, 2, ...,Q]$
>         * **while** not done **do**
>             * Sample mini-batch $S$
>             * **for** $m$ in $[1, 2, ...,M]$
>                 * Randomly sample a class pair $(i, j)$
>                 * Obtain the augmented set $S'$ by corrupting labels $(i, j)$ in $S$ (Eq. 1)
>                 * Derive the augmented model $\theta'_m(w)$ by applying weighted version of single-step gradient descent on theta using $S'$ (Eq. 8)
>                 * Derive the augmented model $\theta'_0(w)$ by applying weighted version of single-step gradient descent on theta using $S$ (Eq. 8)
>             * Compute the $L_{RW}$ (Eq. 9)
>             * Update $\{w_n\}$ by minimizing $L_{RW}$

---

> ### Author Response · Authors · 2022-11-18
> **Response to Reviewer szQw (Part 1)**
>
> We sincerely thank the reviewer for the constructive comments and critical suggestions, which fundamentally help us strengthen our work. We do our best to address the concerns raised by all ***seven*** reviewers assigned to our submission. Please see below for our responses and clarifications.
>
> **Q1: I’m not fully understand the sentence “As for instance-level pretext tasks (e.g., rotation prediction or contrastive-based instance discrimination), they are expected to produce proper representations from unlabeled data; it is not clear whether such techniques would result in robust representations when handling input data with noisy labels” in the introduction. In fact, the pretext tasks usually do not depend on labels, so I’m not clear why the data with noisy labels might affect the model output aided by this technique. I cannot see the connections between these two things.**
>
> A1: We really appreciate the reviewer for giving us this opportunity to clarify this statement.
>
> As noted in the 3rd paragraph of Sect. 1, exiting SSL works typically perform data augmentation and derive the associated pretext tasks for training learning models, without access to any label supervision. As we pointed out in the same paragraph, these SSL techniques only perform data augmentation or manipulation at the instance level, and such pretext tasks are not associated with the noisy label learning (NLL) problem.
>
> To address this concern, we propose set-level self-supervision learning (SLSSL) on noisy label data. In our SLSSL, a novel data augmentation strategy is introduced at the set level via label corruptions, so that self-supervision can be observed directly from noisily-labeled training data as the pretext task during training. Specifically, our SLSSL advances label corruption for each mini-batch during training, followed by enforcing the prediction consistency across the resulting models derived by single-step gradient descent using the label-corrupted mini-batches, encouraging the learned model to exhibit sufficient robustness against label noises. This learning strategy can be viewed as a meta-learning technique introduced in MAML [A], which also imposes learning objectives on the single-step updated model for fast model adaptation.
>
> To further verify the advantages of set-level SSL over instance-level SSL approaches to NLL tasks, we conduct additional experiments and compare our SLSSL to recent instance-level SLL approaches MOIT [B] on CIFAR-10. For fair comparison, we follow MOIT to adapt a single PreAct ResNet-18 network, and report the test accuracy in the last training epoch. As can be seen from the table, our SLSSL reached comparable performance for assymetric noise at 40%, while outperformed MOIT with significant margins for different symmetric noise levels.
>
>
> |                     | Sym 20% | Sym 80% | Asym 40% |   Mean    |
> |:------------------- |:-------:| ------- | -------- |:---------:|
> | MOIT [B] (reported) |  94.08  | 75.83   | 93.27    |   87.73   |
> | Our SLSSL           |  95.18  | 92.82   | 92.23    | **93.41** |
>
> Based on the above explanations, we have rephrased the second part of the sentence from “it is not clear whether such techniques would result in robust representations when handling input data with noisy labels.” to “it is not clear whether such techniques would result in robust representations when *tackling the problem of noisy label learning (NLL)*”.

---

### Official Review · Reviewer_yAaK · 2022-10-27

**Confidence:** 3
**Correctness:** 2
**Technical Novelty And Significance:** 2
**Empirical Novelty And Significance:** 2
**Recommendation:** 3

**Clarity, Quality, Novelty And Reproducibility:**

This paper lacks reasonable motivation.

Why EMA and labels corrupt are used without a clear explanation?

The connection between corrupting the labels and model robustness remains to be explained. In my opinion, label corrupt may introduce unnecessary noise and confuse the model even more.
How is label corrupt carried out in the experiment?

No code is provided in this paper, and innovation is limited.


**Strength And Weaknesses:**


Strength:

This paper tries to use set-level self-supervised learning (SLSSL) to address noisy label learning (NLL) tasks.

The proposed SLSSL can be utilized for sample reweighting technique based on a novel high-impact principle.



Weakness:

The paper writing is a bit confusing, for example, “additional supervisory signal can be derived to improve robustness of the model against label noise.” and “it is not clear whether such techniques would result in robust representations when handling input data with noisy labels.” These two points contradict each other.

This paper lacks sufficient theoretical support. Can you provide a theoretical analysis of this method regarding stability and generalizability?

Experiments should be conducted on more and larger data sets.


**Summary Of The Paper:**

This paper focuses on the task of noisy label learning (NLL), which proposes set-level self-supervised learning (SLSSL) to model noisy label data.

SLSSL performs self-supervised learning at mini-batch levels with observed noisy labels.

The proposed method relabel the samples from label i using label j, which is called label corrupt, an augmentation method.


**Summary Of The Review:**


The contribution of this paper is limited and the paper lacks a reasonable explanation of the proposed modules.

It is suggested to explain the motivation of this paper in detail.

---

> ### Author Response · Authors · 2022-11-18
> **Response to Reviewer yAaK (Part 5)**
>
> **Q6: The connection between corrupting the labels and model robustness remains to be explained. In my opinion, label corrupt may introduce unnecessary noise and confuse the model even more. How is label corrupt carried out in the experiment?**
>
> A6: We thank the reviewer for raising up this concern. We are more than happy to clarify this issue. More importantly, we will explain why such label manipulation scheme can be viewed as a meta-learning strategy for learning robust NLL models.
>
> We now present the intuition and connection between label corruption and model robustness. As discussed in Sect. 3.2, the label-corrupted set is used to update the learning model by a single-step gradient descent to obtain an *augmented* model, on which a prediction consistency constraint is enforced to encourage the learning model to exhibit robustness against such label corruption. As also noted in the above Q2 and Q5, this learning strategy can be viewed as a meta-learning technique introduced in MAML [J], which advocates the single-step optimization for *model adaptation*. With such a meta-learning strategy and consistency constraint, the learned model would be expected to exhibit sufficient robusteness when training from the entire noisy label training set.
>
> **Q7: No code is provided in this paper, and innovation is limited.**
>
> A7: We thank the reviewer for the critical concern and suggestion. We will be glad to release the source code (with possible revision and additional experiments suggested by all seven reviewers). As for novelty, please kindly refer to Q5 and Q6 which explains how our proposed set-level self-supervised learning scheme allows training of robust NLL models. From the experiment results in Sect. 4 (and responses in Q1 and Q3) on four datasets (CIFAR-10/10N/100 and Clothing1M) compared to MLNT [I], DivideMix [K], and MOIT [D], our proposed SLSSL has been shown to perform favorably against baseline and SOTA methods of NLL and SSL.
>
> **Rerefence**
> [A] D. Hendrycks et al., Using self-supervised learning can improve model robustness and uncertainty, NeurIPS 2019
> [B] A. Ghosh et al., Contrastive learning improves model robustness under label noise, CVPRW, 2021
> [C] Y. Yao et al., Jo-src: A contrastive approach for combating noisy labels, CVPR 2021
> [D] D. Ortego et al., Multi-objective interpolation training for robustness to label noise, CVPR 2021
> [E] T. Chen et al., A Simple Framework for Contrastive Learning of Visual Representations, ICML 2020
> [F] K. He et al., Momentum Contrast for Unsupervised Visual Representation Learning, CVPR 2019
> [G] J.-B. Grill t al., Bootstrap Your Own Latent: A New Approach to Self-Supervised Learning, NeurIPS 2020
> [H] N. Saunshi et al., A theoretical analysis of contrastive unsupervised representation learning, ICML 2019
> [I] J. Li et al., Learning to learn from noisy labeled data, CVPR 2019
> [J] C. Finn et al., Model-agnostic meta-learning for fast adaptation of deep networks, ICML 2017
> [K] J. Li et al., Dividemix: Learning with noisy labels as semi-supervised learning, ICLR 2020
> [L] E. Arazo et al., Unsupervised Label Noise Modeling and Loss Correction, ICML 2019
> [M] H. Wang et al., ProMix: Combating Label Noise via Maximizing Clean Sample Utility, IJCAI-ECAI 2022
> [N] H. Song et al., Learning from noisy labels with deep neural networks: A survey, TNNLS 2021
> [O] G. Patrini et al., Making deep neural networks robust to label noise: a loss correction approach, CVPR 2017
> [P] D. Hendrycks et al., Using trusted data to train deep networks on labels corrupted by severe noise, NeurIPS 2018
> [Q] X. Xia et al., Are anchor points really indispensable in label-noise learning?, NeurIPS 2019
> [R] Z. Wang et al., Training noise-robust deep neural networks via meta-learning, CVPR 2020
> [S] Y. Yao et al., Dual-T: Reducing estimation error for transition matrix in label-noise learning, NeurIPS 2020
> [T] A. Tarvainen et al., Mean teachers are better role models: Weight-averaged consistency targets improve semi-supervised deep learning results, NeurIPS 2017
> [U] D. T. Nguyen et al., SELF: Learning to Filter Noisy Labels with Self-Ensembling, ICLR 2020
> [V] T. Zhou et al., Robust curriculum learning: From clean label detection to noisy label self-correction, ICLR 2021

---

> > ### Comment · Reviewer_yAaK · 2022-11-24
> > **Responses to Authors**
> >
> > Thanks for your responses. The authors addressed part of my concerns in their response. I keep my rating.

---

> ### Author Response · Authors · 2022-11-18
> **Response to Reviewer yAaK (Part 4)**
>
> **Q4: This paper lacks reasonable motivation.**
>
> A4: We thank the reviewer for raising this concern. We would appreciate if the reviewer would clarify that the motivation of interest is the need of solving NLL, or the design of our proposed SLSSL scheme.
>
> For noisy label learning (NLL), it is among the active research problems in machine learning communities [N, O, P, Q, R, S, K]. As noted in the 2nd and 3rd paragraphs of Sect. 1, existing NLL approaches [O, P, Q, R, S, K] rely on accurate estimation of the noise transition matrix or sample weights, which require proper learning and estimation using the training data and their noisy labels. Also, existing self-supervised learning (SSL) based NLL approaches [A, B, C, D] consider pretext tasks at the instance level only, which is not associated with the task of NLL.
>
> As for the proposed SLSSL, we propose a data/label augmentation strategy. This unique scheme is conducted at the set level via label corruption, so that self-supervision guidance can be designed as pretext tasks for training NLL models. Such a proposed strategy further allows our SLSSL to produce additional self-supervisory signals to estimate the noise transition matrix and to design sample reweighting algorithms for further performance improvement. Finally, from our experiments in Sect. 4 (and also responses in Q1 and Q3), our SLSSL was shown to perform favorably against SOTA SSL and NLL methods [D, I, K, L]. Thus, the effectiveness and robustness of our proposed learning scheme can be successfully verified.
>
> **Q5: Why EMA and labels corrupt are used without a clear explanation?**
>
> A5: We thank the reviewer for raising up the concerns on the use of EMA and our label corruption for providing set-level self-supervision. We now address these two concerns below.
>
> We feel that it is necessary to first explain label corruption, which is conducted for each mini-batch and serves as the core design of our SLSSL. Given a set (i.e., a mini-batch) of training samples S with noisy labels, we propose to corrupt the labels in S during training, which provides a unique way to *augment* the training set for mimicking the noisily-labeled training data. For each mini-batch, we randomly sample a class pair to perform label swap as label corruption, and the prediction consistency on the remaining class labels can be expected. This is how we design our pretext tasks with self-supervisory guidance to train NLL models. Specifically, we enforce the prediction consistency across the resulting models derived by single-step gradient descent using the label-corrupted mini-batches. As pointed out in Q2, this learning strategy can be viewed as a meta-learning technique introduced in MAML [J], which advocates the single-step optimization for model learning and adaptation. The above remarks can be found in the 1st paragraph of Sect. 3.3. Also, as discussed in Q1, existing instance-level SSL techniques (e.g., contrastive-based instance discrimination such as those in Jo-SRC [C] and MOIT [D]) cannot be directly applied to NLL. This is confirmed by the additional experiments presented in Q1.
>
> On the other hand, EMA (exponential moving average) is a common technique in training deep neural networks for alleviate possible overfitting problems, which has been widely applied in SSL [F] and NLL [I, U, V] works. Thus, we also utilize the EMA strategy to derive a *stable* NLL model, which is gradually updated by each single-step optimized model (observing performance consistency across mini-batches with different label corruption).
>
> We hope that the above explanations would be sufficient to clarify the raised concerns.

---

> ### Author Response · Authors · 2022-11-18
> **Response to Reviewer yAaK (Part 3)**
>
> **Q3: Experiments should be conducted on more and larger data sets.**
>
> A3: We thank the reviewer for the suggestion, and we are happy to include more experiments on additional datasets to alleviate the raised concern.
>
> Due to limited time and computing resources for rebuttal, we select two noise types (20% and 50% symmetric noise) on CIFAR-100, and we report results using the same random seed to ensure the same noise setting between DivideMix [K] and our SLSSL. We follow DivideMix and report the best test accuracy (Best) and the averaged test accuracy over the last 10 epochs (Last). From the results shown in the table below, improved performances over the SOTAs of DivideMix and MD-DYR-SH [L] were achieved.
>
>
> |                                 |                | Sym 20% | Sym 50% |
> |:------------------------------- |:--------------:|:------------------------:|:------------------------:|
> | MD-DYR-SH [L] (reported)   | Best  |      73.9       |      66.1       |
> | MD-DYR-SH [L] (reported)   |  Last |       73.4      |      65.4      |
> | DivideMix [K] (reproduced) | Best  |     77.50      |     74.20      |
> | DivideMix [K] (reproduced) |  Last |      77.00     |     73.80     |
> | Our SLSSL                       | Best  |   **78.08**   |   **74.34**   |
> | Our SLSSL                       | Last |   **77.83**   |   **73.95**   |
>
> With the above CIFAR-100 dataset, a total of four datasets have been considered in our work. We note that, use of 2 or 3 datasets for evaluating NLL works has been seen in recent literatures (e.g., MLNT [I] considered CIFAR-10 and Clothing1M, and ProMix [M] considered CIFAR-10N and CIFAR-100N). We hope the reviewer would agree that, based on our evaluation of CIFAR-10, CIFAR-10N, CIFAR-100, and Clothing1M, the effectiveness and robustness of our work can be sufficiently verified.

---

> ### Author Response · Authors · 2022-11-18
> **Response to Reviewer yAaK (Part 2)**
>
> **Q2: This paper lacks sufficient theoretical support. Can you provide a theoretical analysis of this method regarding stability and generalizability?**
>
> A2: We thank the reviewer for giving us the opportunity to provide additional justification of our work.
>
> First of all, self-supervised learning (SSL) is referring to the learning strategy that, without label supervision, one can manipulate data samples and use the associated guidance to train the learning model. For example, in SimCLR [E], two types of data augmentation are applied to an input image to obtain two views, which are regarded as a positive data pair to derive the contrastive loss as the self-supervisory signal. While SSL has been shown to introduce excellent representaiton learning ability for neural network (e.g., SimCLR [E], MoCo [F], and BYOL [G]), its theoretical foundations can be found in recent iterature like [H].
>
> We now provide further explanations and supports to why the proposed set-level SSL can be applied for noisy-label learning (NLL) problems. As pointed out in MLNT [I], the main issue for training a model directly on noisily-labeled data is that it tends to overfit to the noisy labels and thus yields poor generalization. One key approach to address this issue is to optimize the model’s parameters toward the direction of less prone to overfitting and more robust against label noise. While instance-based SSL-based approaches to NLL like Jo-SOC [C] and MOIT [D] have been proposed, such SSL techniques did not utilize the observed noisy labels in their pretext learning stage. Our SLSSL advances label corruption for each mini-batch during training, followed by enforcing the prediction consistency across the resulting models derived by single-step gradient descent using the label-corrupted mini-batches, encouraging the learned model to exhibit sufficient robustness against label noises. This learning strategy can be viewed as a meta-learning technique as introduced in *MAML* [J]. With such single-step optimization for model learning and adaptation has been fundamentally justified in MAML, we hope that the above justification would strenthen the design of our proposed SLSSL scheme.
>
> It is also worth noting that, as noted at the end of Sect. 3.2, our proposed SLSSL strategy can be utilized for model training (Sect. 3.3) and sample reweighting (Sect. 3.4), allowing us to perform EM-like iterative learning when training NLL models (Sect. 3.5). This also brings a connection between the use of SLSSL and the EM algorithm. We take the K-means clustering algorithm as an example, and compare it with our EM-like iterative learning strategy in the following table.
>
>
> |            | K-means                          | SLSSL |
> |:---------- |:-------------------------------- |:----- |
> | 1st E-step | Randomly initialize K centroids. |   Randomly initialize model parameters, and train the model against noisy labels using samples (with equal weights).    |
> | M-step     |  Determine cluster membership for each sample based on the K centroids.                                |   Determine the weights for all samples based on the trained model.    |
> | E-step     | Re-compute K centroids based on the updated cluster membership.                             | Re-train the model based on the updated sample weights.  |
>
> As can be seen from the table, our model training and sample reweighting algorithms can be combined to form an EM-like algorithm. Furthermore, the effectiveness of this EM-like iterative learning strategy was verified by Figure 7 in the appendix.

---

> ### Author Response · Authors · 2022-11-18
> **Response to Reviewer yAaK (Part 1)**
>
> We sincerely thank the reviewer for the constructive comments and critical suggestions, which fundamentally help us strengthen our work. We do our best to address the concerns raised by all ***seven*** reviewers assigned to our submission. Please see below for our responses and clarifications.
>
> **Q1: The paper writing is a bit confusing, for example, “additional supervisory signal can be derived to improve robustness of the model against label noise.” and “it is not clear whether such techniques would result in robust representations when handling input data with noisy labels.” These two points contradict each other.**
>
> A1: We thank the reviewer for raising up possible confusion in the introduction section. We are happy to address this issue and to clarify the concern.
>
> The two statements are applied to describe the *property* of existing instance-based self-supervised learning (SSL) approaches and their *limitation* to NLL tasks, respectively. More precisely, the former statement refers to the use of data augmentation or manipulation [A, B, C, D], which allows auxiliary supervision as pretext tasks for training learning models. As for the latter statement, it points out that existing SSL techniques typically perform the above augmentation or manipulation at the instance level, which is not associated with the task of interest (i.e., noisy label learning, NLL).
>
> To further verify the latter point, we add an additional experimental comparison of our SLSSL with an instance-level SSL-based NLL method MOIT [D] on CIFAR-10. For fair comparison, we follow MOIT to adapt a single PreAct ResNet-18 network, and report the test accuracy in the last training epoch. As can be seen from the table, our SLSSL reached comparable performance for assymetric noise at 40%, while outperformed MOIT with significant margins for different symmetric noise levels. We confirm that the use of existing instance-level SSL techniques would not be sufficient for training NLL models, while our proposed set-level self-supervised learning scheme would achieve satisfactory performances.
>
>
> |                     | Sym 20% | Sym 80% | Asym 40% |   Mean    |
> |:------------------- |:-------:| ------- | -------- |:---------:|
> | MOIT [D] (reported) |  94.08  | 75.83   | 93.27    |   87.73   |
> | Our SLSSL           |  95.18  | 92.82   | 92.23    | **93.41** |
>
> From the above explanation, we hope the reviewer can see that these two statements are actually not contradicting with each other. To alleviate possible confusion, we have rephrased the second part of the sentence from “it is not clear whether such techniques would result in robust representations when handling input data with noisy labels.” to “it is not clear whether such techniques would result in robust representations when *tackling the problem of noisy label learning (NLL)*”.

---

### Official Review · Reviewer_rqJk · 2022-10-27

**Confidence:** 3
**Clarity, Quality, Novelty And Reproducibility:** 1) Other than the set-level SSL appro…
**Correctness:** 3
**Technical Novelty And Significance:** 3
**Empirical Novelty And Significance:** 2
**Recommendation:** 5

**Strength And Weaknesses:**


Strengths:

1) The paper is clearly written and does a great job at explaining the proposed method in details.

2) The proposed method is interesting and introduces the notion of set-level self-supervised learning, and invariance to label corruption, which is a new concept in the field of learning with noisy labels.


Weaknesses:


1) Calling the method self-supervised feels odd given that the task to solve is (noisily) supervised in itself, and that the gradients from the $\mathcal{L}_{CE}$ loss are retro-propagated into $\theta$.

2) How does the model perform without $\mathcal{L}_{CE}$ ? In particular, it would be interesting to train the model with only with the self-supervised loss and evaluate the model on a linear frozen classification task, similar to what is done in standard SSL [1].

3) How do you balance $\mathcal{L}_{CE}$ and the self-supervised loss ? More generally, it would be great to have an ablation on the different criterion of the method, in order to better understand the role that each one plays.

4) The experimental results are a bit weak. Comparison with several recent methods cited in the paper are missing. In particular none of the methods mentioned in the "Self-Supervised Learning for NLL" paragraph of the related work section are compared in the experiences. For exemple, Jo-SRC [2] achieves 75.93\% test accuracy on the clothing-1M dataset against 74.51\% reported by SLSSL in the paper.

5) It would be great to have results on the Cifar-100 dataset which has significantly more classes than Cifar-10 and Clothing-1M.

Remarks and questions:

1) Do you apply different augmentations (Random cropping, color jittering) in the batches S and S' ?.

2) "most SSL strategies are performed at the instance level, regardless of the correctness of its label". I don't understand this sentence, as there is no label in SSL.

3) The parallel made with the EM algorithm feels a bit far-fetched, would it be possible to provide a more mathematical comparison ?

[1] T. Chen et al., A Simple Framework for Contrastive Learning of Visual Representations, ICML 2020
[2] Y. Yao et al., Jo-SRC: A Contrastive Approach for Combating Noisy Labels, CVPR 2021

**Summary Of The Paper:**

This paper tackles the problem of learning with noisy-labels. By combining different existing approaches, such as noise transition matrix estimation and sample reweighing, with a new set-level self-supervised learning pipeline, the proposed approach achieve strong performance on NNL (noisy label learning) vision benchmarks. The method can also be seen as an expectation-maximization (EM) algorithm.

**Summary Of The Review:**

The proposed method is conceptually interesting and valuable for the community. However it is hard to understand the contribution of all the components of the proposed method and the results are not significantly better compared to the current state-of-the-art. The paper would be greatly improved if the benefit of using a self-supervised component for learning with noisy-labels was clearer.

---

> ### Author Response · Authors · 2022-11-18
> **Response to Reviewer rqJk (Part 4)**
>
> **Q10: "accurately estimate the noise transition matrix remains a challenging problem" -> This is still a challenge for the proposed method. More generally, could you explain how your method alleviate the short-comings of related works?**
>
> A10: We thank the reviewer for giving us this opportunity to further clarify this point. Several NLL approaches require an additional clean training subset for estimating the noise transition matrix [H, I]. We alleviate such short-comings by removing the above training requirement and proposing a unique set-level self-supervised learning scheme for NLL. We agree that, without the above training requirement, it is an even *more* challenging problem of estimating the noise transition matrix. However, when comparing to recent NLL methods on Clothing 1M, our SLSSL (74.51%) was observed to perform against methods of F-correction [J] (69.84%), MLC [I] (71.06%), and T-Revision [K] (74.18%), in which T-Revision also estimates the noise transition matrix without observing clean training data subset. These experimental results support the effectiveness and robustness of our proposed learning scheme.
>
> **Q11: Figure 1 is unclear. It is hard to understand the intuition of the method by watching at the Figure, as it feels like a single representation and class probability vector are computed for the entire set. This might also be due to the naming "set-level SSL" which sounds like an SSL loss is applied at the same level and not at the instance level.**
>
> A11: We thank the reviewer for the critical suggestion. Originally, Figure 1 is used to illustrate the difference between instance-level SSL and our proposed set-level SSL. While instance-level SSL focuses on maximizing *feature* agreement between two augmentations of the same *instance*, our SLSSL focuses on maximizing the *performance* agreement between two augmented models obtained from different label corruptions of the same *set*.
>
> Based on the remarks provided, we will edit Figure 1 by replacing the 4 circles at the model output under the SLSSL by 4 probability distributions, each indicating a prediction vector for a single input image. Also, the text “similarity” is replaced by “consistency” to align with our terminology used in Sect. 3. We thank the reviewer again for the very helpful suggestion.
>
>
> **Q12: The difference between "single model" and "co-training model" needs to be clarified, the term "single model" only appears in Section 4.2 for the first time. What is the difference between Table 1 and Table 2 ? And between "w/o co-training" and "w/ co-training" in Table 2?**
>
> A12: We thank the reviewer for pointing this out, and we are happy to further clarify this issue.
>
> It is correct that the terms “single model” and “w/o co-training” denote the use of a single neural network, while “co-training model” and “w/ co-training” represent the approaches involving dual models that are jointly trained. Table 1 is to compare the NLL performances of approaches using a single model, while Table 2 is for comparisons of dual-model NLL methods (e.g., DivideMix [F]). Tables 1 and 2 can also be viewed as ablation studies, showing deployment of dual models with properly designed learning strategies can further improve the robustness and effectiveness of NLL models. We thank the reviewer again for the suggestion, and we will use the terms single/dual models in our revised version for better understanding.
>
>
>
> **Rerefence**
> [A] Y. Yao et al., Jo-src: A contrastive approach for combating noisy labels, CVPR 2021
> [B] D. Ortego et al., Multi-objective interpolation training for robustness to label noise, CVPR 2021
> [C] T. Chen et al., A Simple Framework for Contrastive Learning of Visual Representations, ICML 2020
> [D] J. Li et al., Learning to learn from noisy labeled data, CVPR 2019
> [E] S. Liu et al., Robust Training under Label Noise by Over-parameterization, ICML 2022
> [F] J. Li et al., Dividemix: Learning with noisy labels as semi-supervised learning, ICLR 2020
> [G] E. Arazo et al., Unsupervised Label Noise Modeling and Loss Correction, ICML 2019
> [H] D. Hendrycks et al., Using trusted data to train deep networks on labels corrupted by severe noise, NeurIPS 2018
> [I] Z. Wang et al., Training noise-robust deep neural networks via meta-learning, CVPR 2020
> [J] G. Patrini et al., Making deep neural networks robust to label noise: a loss correction approach, CVPR 2017
> [K] X. Xia et al., Are anchor points really indispensable in label-noise learning?, NeurIPS 2019

---

> > ### Comment · Reviewer_rqJk · 2022-11-24
> > **Responses to authors**
> >
> > Thanks for your answers. I still have concerns regarding the overall performance of the method and I will therefore keep my rating.

---

> ### Author Response · Authors · 2022-11-18
> **Response to Reviewer rqJk (Part 3)**
>
> **Q6: Do you apply different augmentations (Random cropping, color jittering) in the batches S and S'?**
>
> A6: No, in our proposed scheme, we perform set-level label corruption to augment data with different types/degrees of label corruption, which aligns with the NLL learning problem and is applied to each mini-batch instead of each image. We did not apply different instance-level data augmentations between S and S’. While it is possible to do so, it would not be easy to determine whether the learned model robustness is toward label corruption or data augmentation. However, we agree that this is an interesting point and will keep this among our future research directions.
>
> **Q7: "most SSL strategies are performed at the instance level, regardless of the correctness of its label". I don't understand this sentence, as there is no label in SSL.**
>
> A7: We thank the reviewer for pointing this out. We understand that existing SSL techniques are generally designed to perform instance-level data manipulation for designing training objectives, without taking any label supervision into consideration.
>
> As for our SLSSL, we uniquely perform label manipulation for samples in each mini-batch, which allows us to directly exploit *noisy* labels from training data for designing training objectives. For simplicity and clarity, we replace the sentence “regardless of the correctness of its label” by “without accessing the label of each instance” in our revised version.
>
> **Q8: The parallel mode with the EM algorithm feels a bit far-fetched, would it be possible to provide a more mathematical comparison?**
>
> A8: We thank the reviewer for raising up this issue. As noted at the end of Sect. 3.2, our proposed SLSSL strategy can be utilized for model training (Sect. 3.3) and sample reweighting (Sect. 3.4), allowing us to perform EM-like iterative learning when training NLL models.
>
> To give a more intuitive mathematical comparison, we take the K-means clustering algorithm as an example, and compare it with our EM-like iterative learning strategy in the following table.
>
>
>
> |            | K-means                          | SLSSL |
> |:---------- |:-------------------------------- |:----- |
> | 1st E-step | Randomly initialize K centroids. |   Randomly initialize model parameters, and train the model against noisy labels using samples (with equal weights).    |
> | M-step     |  Determine cluster membership for each sample based on the K centroids.                                |   Determine the weights for all samples based on the trained model.    |
> | E-step     | Re-compute K centroids based on the updated cluster membership.                             | Re-train the model based on the updated sample weights.  |
>
> As can be seen from the table, our model training and sample reweighting algorithms can be combined to form an EM-like algorithm. Furthermore, the effectiveness of this EM-like iterative learning strategy was verified by Figure 7 in the appendix. It can be observed from Figure 7 that, as the sample weights being updated with the increase of M-step iterations (i.e., increased AUC for clean/noisy sample classification in Figure 7a), the model performance also got improved with the increase of E-step training epochs (i.e., increased test accuracy in Figure 7b), and *vice versa*. For example, the test accuracy jumped from 86.5% to 89.6% after the first round of M-step that results in an AUC around 0.82. Similarly, the AUC values were gradually improved from 0.82 to 0.85 after two rounds of sample reweighting iterations, in which a stronger model (test acc: 86.5% to 89.6%) is utilized in the M-steps. We hope that the above clarification would sufficiently address the reviewer’s concern.
>
> **Q9: "MLNT corrupts labels randomly" -> This is also the case in the procedure described in Section 3.2**
>
> A9: We thank the reviewer for pointing this out. Indeed, both MLNT [D] and our SLSSL corrupt labels within a mini-batch randomly. MLNT simply enforces performance consistency between the models learned with different random label corruption. However, we randomly corrupt the labels in a more structural way. That is, we randomy select two categories from each mini-batch for label corruption (i.e., label swap). This allows our SLSSL to produce additional supervisory signal to (i) estimate the noise transition matrix and (ii) design sample reweighting algorithms to train NLL models.

---

> ### Author Response · Authors · 2022-11-18
> **Response to Reviewer rqJk (Part 2)**
>
> **Q3: How do you balance $L_{CE}$ and the self-supervised loss? More generally, it would be great to have an ablation on the different criterion of the method, in order to better understand the role that each one plays.**
>
> A3: We thank the reviewer for pointing out this practical issue. In our implementation, we follow MLNT [D] (also performing data augmentation for each mini-batch with meta-learning strategy) and set the learning rate of the classification loss $L_{CE}$ and the SSL loss in a 1:2 ratio. Due to limited time and computing resources during rebuttal, we choose to conduct experiments and comparisons with SOP [E], DivideMix [F], and MOIT [B] on CIFAR-10, CIFAR-10N, and CIFAR-100 datasets (as suggested by *all* reviewers). We will be happy to include the suggested ablation study in the future version.
>
> **Q4: The experimental results are a bit weak. Comparison with several recent methods cited in the paper are missing. In particular none of the methods mentioned in the "Self-Supervised Learning for NLL" paragraph of the related work section are compared in the experiences. For exemple, Jo-SRC achieves 75.93% test accuracy on the clothing-1M dataset against 74.51% reported by SLSSL in the paper.**
>
> A4: We thank the reviewer for this critical remark and suggestion. For completeness, we directly copy and add the performance of Jo-SRC [A] on Clothing1M to Table 4. However, since the noisy label settings for Jo-SRC on CIFAR-10 and CIFAR-100 are not identical to those considered in our experiments, plus the official code of Jo-SRC is not publicly available, we are not able to include their results of these two datasets for rebuttal purposes. However, as discussed earlier in Q2, we compared our SLSSL with another SSL-based NLL methods MOIT [B] on CIFAR-10 and observed a significant improvement of our set-level SSL over instance-level ones.
>
> **Q5: It would be great to have results on the Cifar-100 dataset which has significantly more classes than Cifar-10 and Clothing-1M.**
>
> A5: We thank the reviewer for the suggestion. We follow the suggestions and conduct additional experiments on CIFAR-100. Due to limited time and computing resources, we only select two noise types (20% and 50% symmetric noise), and report results from a single run using the same random seed to ensure the same noise setting between DivideMix [F] and our SLSSL. We follow DivideMix and report the best test accuracy (Best) and the averaged test accuracy over the last 10 epochs (Last). From the results shown in the table below, it can be seen that improved performances over the SOTAs of DivideMix and MD-DYR-SH [G] were achieved.
>
>
> |                                 |                | Sym 20% | Sym 50% |
> |:------------------------------- |:--------------:|:------------------------:|:------------------------:|
> | MD-DYR-SH [G] (reported)   | Best  |      73.9       |      66.1       |
> | MD-DYR-SH [G] (reported)   |  Last |       73.4      |      65.4      |
> | DivideMix [F] (reproduced) | Best  |     77.50      |     74.20      |
> | DivideMix [F] (reproduced) |  Last |      77.00     |     73.80     |
> | Our SLSSL                       | Best  |   **78.08**   |   **74.34**   |
> | Our SLSSL                       | Last |   **77.83**   |   **73.95**   |

---

> ### Author Response · Authors · 2022-11-18
> **Response to Reviewer rqJk (Part 1)**
>
> We sincerely thank the reviewer for the constructive comments and critical suggestions, which fundamentally help us strengthen our work. We do our best to address the concerns raised by all ***seven*** reviewers assigned to our submission. Please see below for our responses and clarifications.
>
> **Q1: Calling the method self-supervised feels odd given that the task to solve is (noisily) supervised in itself, and that the gradients from the $L_{CE}$ loss are retro-propagated into $\theta$.**
>
> A1: We thank the reviewer for raising this concern. We understand that self-supervied learning (SSL) techniques generally focus on unlabeled data, and thus it causes possible confusion when proposing SSL-based solutions to noisy label learning (NLL)  problems. We now clarify this issue below.
>
> In SSL, one typically applies instance-level data manipulation without the access to any label information, so that the associated guidance can be observed as training guidance. As for NLL, training data are assigned with noisy labels. To handle such problems, we propose a set-level self-supervised learning (SLSSL) scheme and design pretext tasks via label manipulation for providing self-supervisory signals. Since such pretext tasks are designed introduce model robustness under label corruption (not fitting noisy label outputs directly), we view our method a scheme of self-supervised learning. And, it is worth repeating that, such SSL scheme is uniquely conducted for data/label in each mini-batch.
>
> We would also like to point out that, in recent SSL-based NLL works like Jo-SRC [A] and MOIT [B], classification loss of $L_{CE}$ is still calculated when training their models. Nevertheless, the focus of NLL works stays in the use of noisy labels for providing proper guidance in learning classification models, with the goal of introducing sufficient robustness into such models.
>
> **Q2: How does the model perform without $L_{CE}$? In particular, it would be interesting to train the model with only the self-supervised loss and evaluate the model on a linear frozen classification task, similar to what is done in standard SSL [C].**
>
> A2: We thank the reviewer for suggesting this interesting direction. We now explain the reason why the use of $L_{CE}$ loss is necessary and cannot be removed from the proposed SLSSL scheme.
>
> First of all, following Q1, the pretext task in our proposed SLSSL performs label manipulation for samples in a mini-batch, with the resulting prediction consistency as self-supervisory objectives. More specifically, our SLSSL advances label corruption for each mini-batch during training, followed by enforcing the prediction consistency across the resulting models derived by single-step gradient descent using the label-corrupted mini-batches, encouraging the learned model to exhibit sufficient robustness against label noises. In other words, noisy labels and the classification loss $L_{CE}$ are jointly utilized in our framework.
>
> Secondly, we would like to point out that the learning scenario in standard SSL [C] (i.e., instance-level SSL followed by linear classification) would not be applicable for NLL problems, since the pretext tasks in such SSL works are not utilized the noisy labels presented during training, and they are not designed for downstream NLL tasks either.
>
> Finally, as pointed out in Q1, recent SSL-based approaches to NLL such as Jo-SRC [A] and MOIT [B] also train their model with both self-supervised loss and classification loss ($L_{CE}$) at the same time. We follow this strategy but propose to perform set-level self-supervisory for NLL. To further confirm the advantage of our set-level SSL strategy over standard SSL (i.e., instance-level) ones, we compare our SLSSL with MOIT on CIFAR-10. For fair comparison, we follow MOIT to adapt a single PreAct ResNet-18 network, and report the test accuracy in the last training epoch. As can be seen from the table, our SLSSL reached comparable performance for assymetric noise at 40%, while outperformed MOIT with significant margins for different symmetric noise levels. This additional experiment further confirms the effectiveness of our SLSSL over recent instance-based SSL approaches to NLL.
>
>
> |                     | Sym 20% | Sym 80% | Asym 40% |   Mean    |
> |:------------------- |:-------:| ------- | -------- |:---------:|
> | MOIT [B] (reported) |  94.08  | 75.83   | 93.27    |   87.73   |
> | Our SLSSL           |  95.18  | 92.82   | 92.23    | **93.41** |

---

### Official Review · Reviewer_fPbP · 2022-11-02

**Confidence:** 4
**Correctness:** 3
**Technical Novelty And Significance:** 3
**Empirical Novelty And Significance:** 2
**Recommendation:** 5

**Clarity, Quality, Novelty And Reproducibility:**

The novelty of the paper is ok. The authors give implementation details, but source codes are not attached.

**Strength And Weaknesses:**

Strength:

The paper proposes a novel method. The idea of putting SLSSL into a EM-like algorithm is interesting.

Weakness:

The name of the problem is somewhat confusing. It is suggested to explain the difference between set-level self-supervised learning and supervised contrastive learning.

SLSSL acts as part of the whole learning strategy. It is advised to give a new name to the whole strategy, which may help readers understand the relation between the whole training process and SLSSL better.

Why should Section 3.2 and Section 3.4 be separated into two parts? Also, the information shown in Fig 2 is all included by Fig 4 and the two figures are very similar. It is suggested to think about the structure of Section 3.

It is advised to explain why SLSSL is only compared with some weak baselines.



**Summary Of The Paper:**

The paper focuses on set-level self-supervised learning (SLSSL) and proposes a new method for this problem with the same name SLSSL. In order to eliminate the bad effects of corrupted labels, SLSSL augments a minibatch by corrupting labels and obtains a augmented model. Then SLSSL enforces the consistency between outputs from the augmented model and the weight-averaged model. The paper then shows how to use SLSSL in model training: the consistency statistics are further translated into noise transition matrix, which refines classification prediction and helps learn the parameters of the model.

**Summary Of The Review:**

The idea is interesting, but the paper still has some problems. It is not qualified right now, but has potential to become better after some revision.

---

> ### Author Response · Authors · 2022-11-18
> **Response to Reviewer fPbP (Part 2)**
>
> **Q3: It is advised to explain why SLSSL is only compared with some weak baselines.**
>
> A3: We thank the reviewer for the valuable suggestion. In our experiments, we considered two weak baselines of F-correction [E] and MLNT [F] in Table 1. Note that F-correction directly estimates the class-wise noise transition matrix based on the classification outputs, and MLNT simply conducts random label corruption for training NLL models. As can be seen from Table 1, our SLSSL achieved significant performance improvements.
>
> As for strong baselines, we considered DivideMix [G] and DM-AugDesc [H] in Tables 2, 3, and 4. For completeness, we now include a SOTA SSL method of MOIT [D], as discussed in Q1. From the results shown in these tables, we see that our method performed favorably against such SOTAs on benchmark datasets of CIFAR-10, CIFAR-10N, and Clothing1M.
>
> **Q4: The authors give implementation details, but source codes are not attached.**
>
> A4: We thank the reviewer for raising this concern. We will be more than happy to release the source code (with possible revision and experiments suggested by all reviewers addressed).
>
> **Rerefence**
> [A] P. Khosla et al.,Supervised Contrastive Learning, NeurIPS, 2020
> [B] T. Chen et al., A Simple Framework for Contrastive Learning of Visual Representations, ICML 2020
> [C] H. Song et al., Learning from noisy labels with deep neural networks: A survey, TNNLS 2021
> [D] D. Ortego et al., Multi-objective interpolation training for robustness to label noise, CVPR 2021
> [E] G. Patrini et al., Making deep neural networks robust to label noise: a loss correction approach, CVPR 2017
> [F] J. Li et al., Learning to learn from noisy labeled data, CVPR 2019
> [G] J. Li et al., Dividemix: Learning with noisy labels as semi-supervised learning, ICLR 2020
> [H] K. Nishi et al., Augmentation strategies for learning with noisy labels, CVPR 2021

---

> ### Author Response · Authors · 2022-11-18
> **Response to Reviewer fPbP (Part 1)**
>
> We sincerely thank the reviewer for the constructive comments and critical suggestions, which fundamentally help us strengthen our work. We do our best to address the concerns raised by all ***seven*** reviewers assigned to our submission. Please see below for our responses and clarifications.
>
> **Q1: The name of the problem is somewhat confusing. It is suggested to explain the difference between set-level self-supervised learning and supervised contrastive learning. SLSSL acts as part of the whole learning strategy. It is advised to give a new name to the whole strategy, which may help readers understand the relation between the whole training process and SLSSL better.**
>
> A1: We thank the reviewer for the suggestion, and we are more than happy to clarify this issue.
>
> *Supervised contrastive learning (SCL)* [A] is a commonly used technique in training deep learning models. Given training data and their labels, one can pull the positive data pairs (i.e., two samples of the same category) to be closer to each other, while repelling the negative ones (i.e., samples of distinct labels) away from each other.
>
> *Self-supervised learning (SSL)* is referring to the learning strategy that, without label supervision, one can manipulate data samples and use the associated guidance to train the learning model. For example, in SimCLR [B], two types of data augmentation are applied to an input image to obtain two views, which are regarded as a positive data pair to derive the contrastive loss as the self-supervisory signal.
>
> As noted above, most SSL techniques are performed at the instance level, while SCL assumes the labels of training data are correct. For noisy label learning (NLL) problems, training data are with noisily annotated labels [C], and one cannot directly apply the above techniques for learning deep neural networks. Thus, we propose set-level self-supervised learning (SLSSL) strategy, which manipulates data samples and their labels in each mini-batch, and enforces the learned model to be robust to such noise corruption. This is also highlighted in our contributions at the end of Sect. 1.
>
>
> To make our discussions and comparisons more complete, we compare our method to MOIT [D], a recent instance-based SSL approach to NLL, on CIFAR-10. For fair comparisons, we follow MOIT to adapt a single PreAct ResNet-18 network, and report the test accuracy in the last training epoch. As can be seen from the table, our SLSSL reached comparable performance for assymetric noise at 40%, while outperformed MOIT with significant margins for different symmetric noise levels.
>
> Based on the above explanation and clarifcation, we hope it becomes clear why it is preferable and necessary for us to use the term noisy label learning (NLL) for describing the task and set-level self-supervised learning (SLSSL) as the proposed learning strategy.
>
> |                     | Sym 20% | Sym 80% | Asym 40% |   Mean    |
> |:------------------- |:-------:| ------- | -------- |:---------:|
> | MOIT [D] (reported) |  94.08  | 75.83   | 93.27    |   87.73   |
> | Our SLSSL           |  95.18  | 92.82   | 92.23    | **93.41** |
>
>
> **Q2: Why should Section 3.2 and Section 3.4 be separated into two parts? Also, the information shown in Fig 2 is all included by Fig 4 and the two figures are very similar. It is suggested to think about the structure of Section 3.**
>
> A2: We are sorry for possible confusion. We understand it is necessary to clarify this issue and to justify the organization of Sect. 3.
>
> With the problem of noisy label learning defined in Sect. 3.1, Sect. 3.2 presents the idea of our proposed set-level self-supervised learning (SLSSL). That is, how we perform set-level label corruption as data manipulation, i.e., label corruption of a given mini-batch with consistency between outputs produced by the associated learned models, so that training strategies and objectives can be applied for solving NLL problems. The learning scheme of SLSSL is illustrated in Figure 2.
>
> Based on Sect. 3.2, Sect. 3.3 details how SLSSL can be applied to solve NLL by training models which are robust and invariant towards noisy label supervision (see Figure 3). As for Sect. 3.4, we explain how SLSSL can be utilized as a sample reweighting technique for NLL (see Figure 4). As concluded in Sect. 3.5, based on Sections 3.3 and 3.4, we further view our SLSSL as an EM algorithm for NLL (i.e., Sect. 3.3 as E-step, and Sect. 3.4 as M-step).
>
>
> We agree that the presentation can be further polished. In addition to adding the above explanation at the end of Sect. 3.1, we suggest to revise the title of Sect. 3.2 as Set-Level Self-Supervised Learning (SLSSL), while maintaining those of Sect. 3.3 and 3.4 as SLSSL for Model Training and SLSSL for Sample Reweighting, respectively.

---

### Official Review · Reviewer_xdaD · 2022-11-03

**Confidence:** 2
**Correctness:** 3
**Technical Novelty And Significance:** 4
**Empirical Novelty And Significance:** 3
**Recommendation:** 5

**Clarity, Quality, Novelty And Reproducibility:**

Generally, the paper is well written. However, I think Fig 2 and 3 confuse the readers.

As the paper mentioned, several methods try to use the SSL task as an auxiliary task to make the main model robust. Can we compare their performance against this paper? If so, please do that.

Fig 5 is not readable in the printed version.

**Strength And Weaknesses:**

+The idea is very interesting. It has been shown that the model efficiently works on noisily labeled data. As far as I know, there has been no similar work previously.
+Results are promising and confirm the feasibility of the idea.

**Summary Of The Paper:**

This paper presents an SSL method to learn from noisy labeled data.  Instead of the previous method, which is based on instance-level SSL, this method is a set level. Authors proposed to learn M models (with parameters $\teta^'_i=(1:M))$ ) wherein in each of the models, the labels of two classes are corrupted. Finally, all models, and also the primary model, are forced to have the same behavior concerning the same inputs.  The final is robust aginst the models and explicitly learning the nosie transition matrix.

**Summary Of The Review:**

The idea is novel, and the paper is well-written.

---

> ### Author Response · Authors · 2022-11-18
> **Response to Reviewer xdaD**
>
> We sincerely thank the reviewer for the constructive comments and critical suggestions, which fundamentally help us strengthen our work. We do our best to address the concerns raised by all ***seven*** reviewers assigned to our submission. Please see below for our responses and clarifications.
>
> **Q1: Generally, the paper is well written. However, I think Fig 2 and 3 confuse the readers.**
>
> A1: We thank the reviewer for raising this concern, and we are more than happy to clarify this issue.
>
> Figure 2 illustrates our proposed set-level supervised learning (SLSSL) framework, depicting how we manipulate data/label at each batch for providing supervision during training. As stated at the end of Sect. 3.2, our SLSSL can be utilized as model training (Sect. 3.3) and sample reweighting (Sect. 3.4) techniques for NLL. Thus, Figure 3 in Sect. 3.3 illustrates how SLSSL is utilized for model training, with SLSSL operation depicted in Figure 2 presented by the “SLSSL” functional block. On the other hand, Figure 4 in Sect. 3.4 depicts how SLSSL can be utilized as a sample reweighting technique for NLL.
>
> We hope the above explanations clarify the purposes of Figures 2, 3, and 4.
>
>
>
> **Q2: As the paper mentioned, several methods try to use the SSL task as an auxiliary task to make the main model robust. Can we compare their performance against this paper? If so, please do that.**
>
> A2: We thank the reviewer for this suggestion, and we are happy to provide more performance comparisons to other SSL-based approaches to noisy label learning.
>
> In the table below, we compare our method with a SOTA SSL-based approach of MOIT [A] on CIFAR-10. For fair comparisons, we follow MOIT to adapt a single PreAct ResNet-18 network, and report the test accuracy in the last training epoch. As can be seen from the table, our SLSSL reached comparable performance for assymetric noise at 40%, while outperformed MOIT with significant margins for different symmetric noise levels.
>
>
> |                     | Sym 20% | Sym 80% | Asym 40% |   Mean    |
> |:------------------- |:-------:| ------- | -------- |:---------:|
> | MOIT [A] (reported) |  94.08  | 75.83   | 93.27    |   87.73   |
> | Our SLSSL           |  95.18  | 92.82   | 92.23    | **93.41** |
>
>
> **Q3: Fig 5 is not readable in the printed version.**
>
> A3: We thank the reviewer for pointing this out, and we agree that the resolution of this figure can be further improved.
>
> The main purpose of Figure 5 is to verify the practicality of the noise transition matrix estimated by our SLSSL. The darker color in each element in Figure 5 indicates a higher value (i.e., close to 1), and lighter colors denote values close to 0. Given the noisy labels and the ground truth label assignments for the training data, the noise transition matrix estimated by our SLSSL (Figure 5b) is expected to be close to the ground-truth one (Figure 5a), which can be visually verified by comparing Figures 5a and 5b.
>
> On the other hand, Figure 5c shows the estimated noise transition matrix *after* a complete sample reweighting process by our SLSSL. With a proper sample reweighting technique applied on NLL data, data with correct labels are expected to be assigned larger weights. In other words, the associated noise transition matrix would be expected to be close to an identity matrix. This can be visually confirmed by Figure 5c, whose diagonal elements are much darker (i.e., close to 1) when comparing to off-diagonal ones (lighter in color and close to 0 in value). We will be happy to adjust the resolution and font so that the figure would be more readable.
>
>
> **Rerefence**
> [A] D. Ortego et al., Multi-objective interpolation training for robustness to label noise, CVPR 2021

---

### Official Review · Reviewer_XcmF · 2022-11-03

**Confidence:** 4
**Clarity, Quality, Novelty And Reproducibility:** The paper is easy to follow and clear…
**Correctness:** 3
**Technical Novelty And Significance:** 2
**Empirical Novelty And Significance:** 2
**Recommendation:** 5

**Strength And Weaknesses:**

Strengths:
1. The paper targets an interesting and relevant problem .
2. The idea of using set-level self-supervised learning  for noisy-label learning is simple and reasonable. The presentation of the proposed method is clear and readable.
3. Experimental results show that the proposed schemes offer gains over the baselines.


Weakness:
1.  Baselines are not enough and the contribution is not clear. There are some existing works which studied the same problems[1, 2, 3]. It would be nice if the authors can discuss the relationships between these existing works and the approach in this paper.   Experimental comparisons with some of these works would be desirable.
2. Experimental results on more datasets and configurations should be included.  Only two datasets are considered for evaluation, which make it hard to estimate the generalisability  and robustness of the proposed approach. The effect of different data augmentation techniques on the proposed approach can also be discussed in the paper or in the  Appendix.
3. Although the authors provide preliminary visualization of the noise transition matrix and the analysis of  sample weights. I think the paper's quality will improve  if the authors can investigate an in-depth  analysis and theoretical justifications on why the use of set-level self-supervised learning improves noisy-label learning
4. The authors do not address the limitations of their work.


[1] Robust Training under Label Noise by Over-parameterization, ICML 2022
[2] ProMix: Combating Label Noise via Maximizing Clean Sample Utility
[3] Beyond Images:Label Noise Transition Matrix Estimation for Tasks with Lower-Quality Features, ICML 2022

**Summary Of The Paper:**

The paper studied how to address the negative impact of noisy labels during training. To tackle this challenge, the authors propose a set-level self-supervised learning loss, which augments a set of images  instead of one instance with label corruption and then maximizes the agreement between two augmented sets.  In addition, the authors show that the proposed learning strategy can also be used for estimating the noise transition matrix and sample selection following typical ways of noisy-label learning.  Experiments and ablation studies demonstrate that the proposed method has better performance than several baselines.



**Summary Of The Review:**

The paper investigated an interesting problem with an easy-to-follow method. But the experiments and more analysis should be provided to demonstrate the idea.

---

> ### Author Response · Authors · 2022-11-18
> **Response to Reviewer XcmF (Part 3)**
>
> **Q5: The novelty could be improved.**
>
> A5: We thank the reviewer for this critical remark. We would like to clarify and highlight the novelty/technical contributions of our method below.
>
> * We propose set-level self-supervised learning (SLSSL) to tackle noisy-label learning (NLL) tasks. Different from most existing SSL works which manipulate data at the instance level, our SLSSL uniquely augment image subsets (i.e., mini-batches) by manipulating their labels, so that self-supervision guidance can be produced accordingly.
> * With aforementioned set-level label manipulation, the proposed self-supervised guidance is designed to enforce the robustness of the learning model. This can be viewed as a pretext task for training models against noisy label data.
> * Without the need of strong statistical assumptions [Q] or collecting an extra set of clean samples [R, S], the above set-level data/label manipulation strategy randomly performs label corruption for a class-pair in a mini-batch. This further allows us to estimate the associated noise transition matrix when training NLL models to counteract the label noise.
> * In addition to training NLL models, our SLSSL can also be utilized to identify the label quality of each training sample, and thus sample reweighting for NLL can be performed accordingly.
> * Fianlly, the proposed SLSSL can be further viewed as an EM-like algorithm, in which the E-step focuses optimizing the model parameters with fixed sample weights, while the M-step focuses on sample reweighting based on the fixed model. As a result, our SLSSL can be uniquely applied as E and M-steps for alternative optimization, which is shown to further improve the NLL performance.
>
>
> **Rerefence**
> [A] S. Liu et al., Robust Training under Label Noise by Over-parameterization, ICML 2022
> [B] H. Wang et al., ProMix: Combating Label Noise via Maximizing Clean Sample Utility, IJCAI-ECAI 2022
> [C] Z. Zhu et al., Beyond Images: Label Noise Transition Matrix Estimation for Tasks with Lower-Quality Features, ICML 2022
> [D] K. Sohn et al., FixMatch: Simplifying Semi-Supervised Learning with Consistency and Confidence, NeurIPS 2020
> [E] Ekin D. Cubuk et al., Randugment: Practical automated data augmentation with a reduced search space, CVPR 2020
> [F] J. Li et al., Dividemix: Learning with noisy labels as semi-supervised learning, ICLR 2020
> [G] D. Berthelot, MixMatch: A Holistic Approach to Semi-Supervised Learning, NeurIPS 2019
> [H] E. Arazo et al., Unsupervised Label Noise Modeling and Loss Correction, ICML 2019
> [I] J. Li et al., Learning to learn from noisy labeled data, CVPR 2019
> [J] T. Chen et al., A Simple Framework for Contrastive Learning of Visual Representations, ICML 2020
> [K] K. He et al., Momentum Contrast for Unsupervised Visual Representation Learning, CVPR 2019
> [L] J.-B. Grill t al., Bootstrap Your Own Latent: A New Approach to Self-Supervised Learning, NeurIPS 2020
> [M] N. Saunshi et al., A theoretical analysis of contrastive unsupervised representation learning, ICML 2019
> [N] Y. Yao et al., Jo-src: A contrastive approach for combating noisy labels, CVPR 2021
> [O] D. Ortego et al., Multi-objective interpolation training for robustness to label noise, CVPR 2021
> [P] C. Finn et al., Model-agnostic meta-learning for fast adaptation of deep networks, ICML 2017
> [Q] G. Patrini et al., Making deep neural networks robust to label noise: a loss correction approach, CVPR 2017
> [R] D. Hendrycks et al., Using trusted data to train deep networks on labels corrupted by severe noise, NeurIPS 2018
> [S] X. Xia et al., Are anchor points really indispensable in label-noise learning?, NeurIPS 2019
> [T] Z. Wang et al., Training noise-robust deep neural networks via meta-learning, CVPR 2020
> [U] Y. Yao et al., Dual-T: Reducing estimation error for transition matrix in label-noise learning, NeurIPS 2020

---

> > ### Comment · Reviewer_XcmF · 2022-11-24
> > **Thanks for your responses**
> >
> > I do appreciate the responses from the Authors. It addresses some of my concerns. However, I still have concerns about the baselines and the technical novelty. I lean to keep my current rating.

---

> ### Author Response · Authors · 2022-11-18
> **Response to Reviewer XcmF (Part 2)**
>
> **Q3: Although the authors provide preliminary visualization of the noise transition matrix and the analysis of sample weights. I think the paper's quality will improve if the authors can investigate an in-depth analysis and theoretical justifications on why the use of set-level self-supervised learning improves noisy-label learning.**
>
> A3: We thank the reviewer for giving us the opportunity to provide additional justification of our work.
>
> Since the key contribution of our work is the proposed set-level self-supervised learning framework (SLSSL), we will start from the supporting remarks on self-supervised learning (SSL). As widely applied for training deep learning models in recent years, SSL is referring to the learning strategy that, without label supervision, one can manipulate data samples and use the associated guidance to train the learning model. For example, in SimCLR [J], two types of data augmentation are applied to an input image to obtain two views, which are regarded as a positive data pair to derive the contrastive loss as the self-supervisory signal. With excellent representation learning ability introduced to the resulting networks (e.g., SimCLR [J], MoCo [K], and BYOL [L]), its theoretical foundations can be found in recent iterature like [M].
>
> As for the problem of interest, i.e., noisy-label learning (NLL) problems, existing SSL works mainly perform instance-level manipulation as self-supervision signals, with are not designed to tackle noisy label data. As pointed out in MLNT [I], the main issue for training a model directly on noisily-labeled data is that it tends to overfit to the noisy labels and thus yields poor generalization. One key approach to address this issue is to optimize the model’s parameters toward the direction of less prone to overfitting and more robust against label noise. While instance-based SSL-based approaches to NLL like Jo-SOC [N] and MOIT [O] have been proposed, such SSL techniques did not utilize the observed noisy labels in their pretext learning stage, as we verified in responses to other reviewers (e.g., Q1 of **Reviewer szQw**). Our SLSSL advances label corruption for each mini-batch during training, followed by enforcing the prediction consistency across the resulting models derived by *single-step gradient descent* using the label-corrupted mini-batches, encouraging the learned model to exhibit sufficient robustness against label noises. This learning strategy can be viewed as a meta-learning technique as introduced in *MAML* [P]. With such single-step optimization for model learning and adaptation has been fundamentally justified in MAML, we hope that the above justification would strenthen the design of our proposed SLSSL scheme.
>
>
> **Q4: The authors do not address the limitations of their work.**
>
> A4: We thank the reviewer for the reminder, and we are happy to discuss the limitations of our proposed work.
>
> Since our proposed SLSSL can be viewed as a unique meta-learning scheme on existing NLL methods like DivideMix [F], we expect longer computation time during training. However, take CIFAR-10 for example, DivideMix took 15 hours to train using a single Nvidia TITAN-V GPU, while the full version of our SLSSL (i.e., implemented as an EM algorithm) required 25 hours. It can be seen that, the overall computation time of our SLSSL is still in the same order of that of SOTAs like DivideMix.
>
> Also, as noted in [Q, R, S, T, U], NLL methods based on class-wise noise transition matrix estimation share the limitation that the number of classes for NLL would be reasonable (e.g., 100 in CIFAR-100). This is to avoid the potential problem of estimating a large noise transition matrix. Sharing the concern of the above works, this would also among the current limitation of our work.

---

> ### Author Response · Authors · 2022-11-18
> **Response to Reviewer XcmF (Part 1)**
>
> We sincerely thank the reviewer for the constructive comments and critical suggestions, which fundamentally help us strengthen our work. We do our best to address the concerns raised by all ***seven*** reviewers assigned to our submission. Please see below for our responses and clarifications.
>
> **Q1: Baselines are not enough and the contribution is not clear. There are some existing works which studied the same problems [A, B, C]. It would be nice if the authors can discuss the relationships between these existing works and the approach in this paper. Experimental comparisons with some of these works would be desirable.**
>
> A1: We thank the reviewer for suggesting three related works. We are happy to include the discussions and comparisons to the suggested works.
>
> SOP [A] proposes to model the label noise and learn to separate it from the data by enforcing its sparsity. ProMix [B] presents a sample selection criterion called matched high-confidence selection (MHCS), which selects samples with high prediction confidence scores as clean data. Similar to our sample reweighting procedure discussed in Sect. 3.4, MHCS can be combined with the small-loss selection criterion to further boost the performance. As for Beyond-Images [C], it is designed to tackle NLL tasks with tabular and text features.
>
> It is worth noting that, ProMix adopted a technique of FixMatch [D], which specifically conducts strong image augmentation such as RandAugment [E] during training for training their NLL model. As for our SLSSL, we follow DivideMix [F] and simply apply *standard* augmentation techniques of random cropping and horizontal flipping. Thus, direct comparisons to ProMix might not be fair without accessing to and revise their code. Also, as noted above, Beyond-Images [C] is mainly designed to tackle tabular and text data, and its use for NLL image classification problems might not be directly applicable. Nevertheless, we follow the suggestion and compare our SLSSL with SOP. Due to limited time and computing resources, we choose CIFAR-10N and Clothing1M for additional evaluation. As listed in the following table, we see that our SLSSL performs favorably against SOP, especially on the more challenging Clothing1M dataset.
>
>
>
>
> |                         | CIFAR-10N (Aggregate) | Clothing1M |
> |:----------------------- |:--------------------------:|:----------:|
> | SOP [A] (reported) |         95.61         |    73.5    |
> | Our SLSSL               |       **95.73**       | **74.51**  |
>
> **Q2: Experimental results on more datasets and configurations should be included. Only two datasets are considered for evaluation, which make it hard to estimate the generalisability and robustness of the proposed approach. The effect of different data augmentation techniques on the proposed approach can also be discussed in the paper or in the Appendix.**
>
> A2: We thank the reviewer for suggesting additional experiments. In addition to the comparisons suggested in Q1, we also conduct experiments on CIFAR-100. Specifically, we select two noise types (20% and 50% symmetric noise) and report results. We follow DivideMix and report the best test accuracy (Best) and the averaged test accuracy over the last 10 epochs (Last). As can be seen from the table below, our SLSSL also outperformed DivideMix and MD-DYR-SH [H] on CIFAR-100.
>
>
> |                                 |                | Sym 20% | Sym 50% |
> |:------------------------------- |:--------------:|:------------------------:|:------------------------:|
> | MD-DYR-SH [H] (reported)   | Best  |      73.9       |      66.1       |
> | MD-DYR-SH [H] (reported)   |  Last |       73.4      |      65.4      |
> | DivideMix [F] (reproduced) | Best  |     77.50      |     74.20      |
> | DivideMix [F] (reproduced) |  Last |      77.00     |     73.80     |
> | Our SLSSL                       | Best  |   **78.08**   |   **74.34**   |
> | Our SLSSL                       | Last |   **77.83**   |   **73.95**   |
>
> It is worth noting that, with the above experiments, a total of *four* datasets are considered in our work. Previous NLL work of MLNT [I] only considered CIFAR-10 and Clothing1M, while ProMix [B] conducted experiments on CIFAR-10N and CIFAR-100N. We hope the reviewer would find that, based on the evaluation of CIFAR-10, CIFAR-10N, CIFAR-100, and Clothing1M, the effectiveness and robustness of our work can be sufficiently verified.

---

### Official Review · Reviewer_HztF · 2022-11-04

**Confidence:** 3
**Correctness:** 4
**Technical Novelty And Significance:** 3
**Empirical Novelty And Significance:** 3
**Recommendation:** 6

**Clarity, Quality, Novelty And Reproducibility:**

The paper is generally well-written. The method part contains multiple steps so an algorithm table would be helpful to understand the overall training process. I think the method is novel. The method seems straightforward and can be reproduced.

**Strength And Weaknesses:**

Strength
1. The writing is clear and the proposed method seems straightforward and easy to implement.
2. The idea of combining self-supervised learning for noisy label learning based on simple intuition is novel.

Weaknesses
1. The improvement over DivideMix seems marginal (Table 2 & 3 & 4). More analysis is needed (statistical test, comparison of training computational cost)
2. The method is tested with one kind of model (ResNet). Can the method generalize to other architectures like Vision Transformers?


**Summary Of The Paper:**

This paper proposes a set-level self-supervised learning method for noisy label learning. The authors propose to enforce model robustness against label corruption by making model predictions (other than the corrupted classes) the same between the original model and augmented models. The noise transition matrix is estimated based on the observation that mislabeled class pairs would lead to fewer differences between the models in the SSL process. The noise matrix is then utilized for sample reweighting. Experiments on two synthetic noisy datasets and one real-world noisy one show the efficacy of the proposed method.


**Summary Of The Review:**

I think the proposed method is clear and novel. Experimental results are a bit weak.

---

> ### Author Response · Authors · 2022-11-18
> **Response to Reviewer HztF (Part 2)**
>
> **Q3: The method part contains multiple steps so an algorithm table would be helpful to understand the overall training process.**
>
> A3: We thank the reviewer for this suggestion. Please see the pseudo code below, and  we will add this algorithm to the appendix.
>
> * Variables
>     * Noisy training dataset: $D$
>     * Learning model: $\theta$
>     * EMA model: $\theta^*$
>     * Number of E-step epochs: $P$
>     * Number of M-step epochs: $Q$
>     * Number of label corruption for each mini-batch: $M$
>     * Estimated noise transition matrix at the $e$-th epoch: $\hat{T}_e$
>     * Estimated noise transition matrix: $\hat{T}$
>
> * E-step (Sect. 3.3)
>     * **if** first E-step
>         * Random initialize $\theta$
>         * Initialize $\theta^*$ by $\theta$
>         * Initialize $\hat{T}$ as identity
>     * **for** $e$ in $[1, 2, ...,P]$
>         * **while** not done **do**
>             * Sample mini-batch $S$ from $D$
>             * **for** $m$ in $[1, 2, ...,M]$
>                 * Randomly sample a class pair $(i, j)$
>                 * Obtain the augmented set $S'$ by corrupting labels $(i, j)$ in $S$ (Eq. 1)
>                 * Derive the augmented model $\theta'_m$ by applying single-step gradient descent on $\theta$ using $S'$ (Eq. 2)
>                 * Compute the KL divergence between $\theta'_m$ and $\theta^*$ (Eq. 4)
>             * Compute set-level self-supervised loss $L_{SSL}$ (Eq. 3)
>             * Update theta by minimizing $L_{SSL}$
>             * Compute classification loss $L_{CE}$ and then correct it using $\hat{T}$ (Eq. 7)
>             * Update theta by minimizing $L_{CE}$
>             * Aggregate all KL divergences derived by corrupting $(i, j)$ to estimate the $(i, j)$-th entry of $\hat{T}_e$ (Eq. 6)
>         * Update $\hat{T}$ using EMA of $\hat{T}_e$
>         * Update $\theta^*$ using EMA of $\theta$
>
> * M-step (Sect. 3.4)
>     * **for** $e$ in $[1, 2, ...,Q]$
>         * **while** not done **do**
>             * Sample mini-batch $S$
>             * **for** $m$ in $[1, 2, ...,M]$
>                 * Randomly sample a class pair $(i, j)$
>                 * Obtain the augmented set $S'$ by corrupting labels $(i, j)$ in $S$ (Eq. 1)
>                 * Derive the augmented model $\theta'_m(w)$ by applying weighted version of single-step gradient descent on theta using $S'$ (Eq. 8)
>                 * Derive the augmented model $\theta'_0(w)$ by applying weighted version of single-step gradient descent on theta using $S$ (Eq. 8)
>             * Compute the $L_{RW}$ (Eq. 9)
>             * Update $\{w_n\}$ by minimizing $L_{RW}$
>
> **Rerefence**
> [A] J. Li et al., Dividemix: Learning with noisy labels as semi-supervised learning, ICLR 2020
> [B] K. Nishi et al., Augmentation strategies for learning with noisy labels, CVPR 2021
> [C] G. Patrini et al., Making deep neural networks robust to label noise: a loss correction approach, CVPR 2017
> [D] J. Li et al., Learning to learn from noisy labeled data, CVPR 2019

---

> ### Author Response · Authors · 2022-11-18
> **Response to Reviewer HztF (Part 1)**
>
> We sincerely thank the reviewer for the constructive comments and critical suggestions, which fundamentally help us strengthen our work. We do our best to address the concerns raised by all ***seven*** reviewers assigned to our submission. Please see below for our responses and clarifications.
>
> **Q1: The improvement over DivideMix seems marginal (Table 2 & 3 & 4). Experiments seems a bit weak. More analysis is needed (statistical test, comparison of training computational cost).**
>
> A1: We thank the reviewer for raising this concern. In Tables 2 and 4, results of DivideMix [A] and DM-AugDesc [B] were directly copied from their papers (i.e., no standard deviation were reported by neither works). To follow the suggestions by the reviewer, we conduct additional runs with different random seeds for our SLSSL and DivideMix on CIFAR-10 and CIFAR-10N. To provide additional performance evaluation, we also add experiments on CIFAR-100. Due to limited time and computing resources during rebuttal, we only select two noise types (20% and 50% symmetric noise) and report results from a single run using the same random seed to ensure the same noise setting between DivideMix and our SLSSL. We report and list the performances in the table below. We note that, following the setting of DivideMix, we report the best test accuracy (Best) and the averaged test accuracy over the last 10 epochs (Last). From this table, it can be seen that the improvements of our SLSSL over DivideMix were statistically significant.
>
>
>
> |                            |      | CIFAR-10 (Asym 40%) | CIFAR-10N (Aggregate) | CIFAR-100 (Sym 20%) | CIFAR-100 (Sym 50%) |
> |:-------------------------- |:----:|:-------------------:|:---------------------:|:-------------------:|:-------------------:|
> | DivideMix [A] (reproduced) | Best |     93.20±0.20      |      95.37±0.12       |        77.50        |        74.20        |
> | DivideMix [A] (reproduced) | Last |     92.35±0.15      |      95.10±0.08       |        77.00        |        73.80        |
> | Our SLSSL                  | Best |   **94.20±0.10**    |    **95.73±0.12**     |      **78.08**      |      **74.34**      |
> | Our SLSSL                  | Last |   **93.75±0.15**    |    **95.57±0.12**     |      **77.83**      |      **73.95**      |
>
> As for training time comparisons, since our proposed SLSSL can be viewed as a meta-learning scheme applied to existing NLL methods like DivideMix, longer computation time can be expected for training the NLL models. However, take CIFAR-10 for example, DivideMix took 15 hours to train using a single Nvidia TITAN-V GPU, while the full version of our SLSSL (i.e., implemented as an EM algorithm) required 25 hours. It can be seen that, the overall computation time of our SLSSL is still in the same order of that of SOTAs like DivideMix. We thank the reviewer again for suggesting additional experiments for evaluation, and we will add the above analysis and comments in the appendix of the revised version.
>
>
> **Q2: The method is tested with one kind of model (ResNet). Can the method generalize to other architectures like Vision Transformers?**
>
>
> A2: Yes, our proposed SLSSL is model agnostic and can be implemented via different network architectures like Vision Transformers. Also, as discussed in Sect. 3 and noted in Q1, our proposed learning scheme can be viewed as a meta-learning strategy, and thus it does not limit to the use of particular network backbones. To provide fair comparisons with SOTA NLL methods, we follow previous works F-correction [C], MLNT [D], DivideMix [A], and DM-AugDesc [B] and apply ResNet as the backbone for experiments. We greatly appreciate the reviewer for pointing out this practical issue, and we hope the above explanation would be sufficient.

---

### Decision · Program_Chairs · 2023-01-20

**Decision:**

Reject

**Justification For Why Not Higher Score:**

Reviewers were concerned about the paper's experiments and lack of baselines.

**Justification For Why Not Lower Score:**

lowest already

**Metareview: Summary, Strengths And Weaknesses:**

Seven experts reviewed the paper, and six of them recommended it "5: marginally below the acceptance threshold" or lower ratings. The common concern was about the paper's experiments and lack of baselines. Hence, the decision is **not** to recommend the paper for acceptance.